# Patchy nanoparticles by atomic stencilling

Ahyoung Kim[1,15], Chansong Kim[1,15], Tommy Waltmann[2,15], Thi Vo[3], Eun Mi Kim[4], Junseok Kim[4], Yu-Tsun Shao[5], Aaron Michelson[6], John R. Crockett[1], Falon C. Kalutantirige[7], Eric Yang[1], Lehan Yao[1], Chu-Yun Hwang[1], Yugang Zhang[6], Yu-Shen Liu[1], Hyosung An[1,8], Zirui Gao[9], Jiyeon Kim[1], Sohini Mandal[7], David A. Muller[5,10], Kristen A. Fichthorn[4,11 ✉], Sharon C. Glotzer[2,3,12 ✉] & Qian Chen[1,7,13,14 ✉]

Stencilling, in which patterns are created by painting over masks, has ubiquitous applications in art, architecture and manufacturing. Modern, top-down microfabrication methods have succeeded in reducing mask sizes to under 10 nm (refs. 1,2), enabling ever smaller microdevices as today's fastest computer chips. Meanwhile, bottom-up masking using chemical bonds or physical interactions has remained largely unexplored, despite its advantages of low cost, solution-processability, scalability and high compatibility with complex, curved and three-dimensional (3D) surfaces[3,4]. Here we report atomic stencilling to make patchy nanoparticles (NPs), using surface-adsorbed iodide submonolayers to create the mask and ligand-mediated grafted polymers onto unmasked regions as 'paint'. We use this approach to synthesize more than 20 different types of NP coated with polymer patches in high yield. Polymer scaling theory and molecular dynamics (MD) simulation show that stencilling, along with the interplay of enthalpic and entropic effects of polymers, generates patchy particle morphologies not reported previously. These polymer-patched NPs self-assemble into extended crystals owing to highly uniform patches, including different non-closely packed superlattices. We propose that atomic stencilling opens new avenues in patterning NPs and other substrates at the nanometre length scale, leading to precise control of their chemistry, reactivity and interactions for a wide range of applications, such as targeted delivery, catalysis, microelectronics, integrated metamaterials and tissue engineering[5–11].

Surface patterning by techniques such as stencilling, embroidery, weaving and marbling has long been used in decorative art, cultural expression and architecture. Among these techniques, stencilling can reproducibly and precisely generate patterns on a surface by applying paint over masks with cut-outs, which is widely used in modern manufacturing. In microfabrication, electronic circuits[12], flow channels[13] and chemical patterns[2] are 'painted' over photoresist masks by lithography at a resolution of tens to hundreds of nanometres, enabling production of miniature devices for electronics[12], memory storage[14] and energy conversion[15]. These masks are fashioned top-down, limiting further feature miniaturization and are often implemented to flat surfaces.

By contrast, stencilling surfaces from bottom-up through self-assembly of atoms, ions or molecules is largely unexplored. Bottom-up processes, which are governed by chemical reactions or physical interactions, are promising in obtaining atomic precision patterning, solution-processability, scalability and compatibility with complex, 3D surfaces. In particular, extensive studies have applied surface patterning to make patchy particles; that is, particles with spatial domains of different ligand chemistries[16–25]. Janus particles, in which a section of a micrometre-sized spherical[16] or ellipsoidal[17] particle is coated with metal and ligand patches, are important examples. However, it remains challenging to synthesize patchy NPs with high precision and high yield for particle sizes less than 100 nm or complex (such as faceted) 3D shapes. This challenge limits the use of patchy NPs as building blocks, for which their surface ligand patterns and resultant directional interaction are expected to have a crucial role in targeted delivery[5], self-assembly into metamaterials[6,7], molecular separation[8] and motored nanomachines[9]. For example, despite various strategies[18–20,26] to render patchy NPs, only their self-assemblies into clusters were demonstrated owing to limited yield. Self-assembly of NPs into extended ordered superlattices, which has been achieved for NPs with polyhedral shapes[27–30], has yet to be realized for patchy NPs.

Here we extend stencilling to a bottom-up platform by combining atomic stencils and polymer paints, providing nanometre precision and uniformity in the surface patterning of differently shaped gold NPs (Fig. 1, Supplementary Figs. 1 and 2 and Supplementary Note 1). At this

[1]Department of Materials Science and Engineering, The Grainger College of Engineering, University of Illinois, Urbana, IL, USA. [2]Department of Physics, University of Michigan, Ann Arbor, MI, USA. [3]Department of Chemical Engineering, University of Michigan, Ann Arbor, MI, USA. [4]Department of Chemical Engineering, The Pennsylvania State University, University Park, PA, USA. [5]School of Applied and Engineering Physics, Cornell University, Ithaca, NY, USA. [6]Center for Functional Nanomaterials, Brookhaven National Laboratory, Upton, NY, USA. [7]Department of Chemistry, University of Illinois, Urbana, IL, USA. [8]Department of Petrochemical Materials Engineering, Chonnam National University, Yeosu, Republic of Korea. [9]National Synchrotron Light Source II, Brookhaven National Laboratory, Upton, NY, USA. [10]Kavli Institute at Cornell for Nanoscale Science, Ithaca, NY, USA. [11]Department of Physics, The Pennsylvania State University, University Park, PA, USA. [12]Biointerfaces Institute, University of Michigan, Ann Arbor, MI, USA. [13]Beckman Institute for Advanced Science and Technology, University of Illinois, Urbana, IL, USA. [14]Materials Research Laboratory, University of Illinois, Urbana, IL, USA. [15]These authors contributed equally: Ahyoung Kim, Chansong Kim, Tommy Waltmann. ✉e-mail: fichthorn@psu.edu; sglotzer@umich.edu; qchen20@illinois.edu

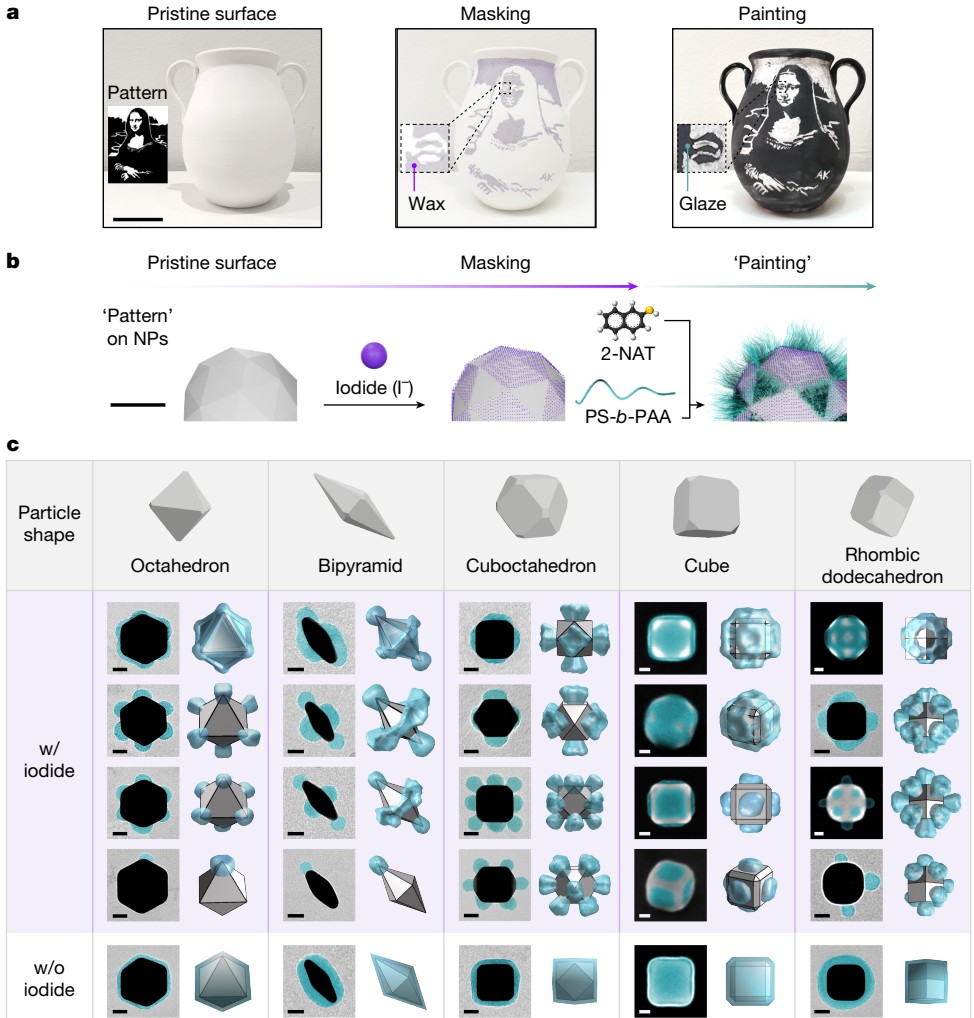

**Fig. 1 | Atomic stencilling generates a library of patch patterns on NPs.**
**a**, Demonstrating stencilling using a Mona Lisa mask on a curved surface of
pottery, which is subsequently painted with ink to create the silhouette.
Mona Lisa pattern artwork adapted from 'Mona Lisa Stencil' by Vincent Robleto,
used with permission. Scale bar, 20 cm. **b**, Schematics of stencilling on faceted
NP surfaces using iodide to create the mask and 2-NAT-mediated PS-*b*-PAA
grafting as paint. Scale bar, 30 nm. **c**, Library of different patchy NPs created by
atomic stencilling of five different NP shapes. The particle shape schematics
also highlight the existence of truncations (not drawn to scale). Within each NP
shape of 'w/ iodide', top to bottom corresponds to increasing [I⁻]/[2-NAT].

Shown for each shape are representative transmission electron microscopy
(TEM) or scanning electron microscopy (SEM) images (left) and predictions
based on polymer scaling theory (right). For cuboctahedra and cubes, two
patch patterns are shown with two different viewing angles. See more examples
in Supplementary Figs. 3–15 and synthesis conditions in Supplementary
Tables 1, 2 and 4–8. In experimental images, the polymer patches are false-
coloured in cyan. The labels 'w/ iodide' (purple-shaded region) and 'w/o iodide'
(unshaded region) refer to polymer-grafted NPs with and without iodide
incubation, respectively. Scale bars, 20 nm.

size scale, we achieve masking and painting by integrating colloidal NP
synthesis and surface science principles with polymer physics. First, we
show that iodides are efficient masks on gold NPs. The co-adsorption
of iodide and 2-naphthalenethiol (2-NAT) on gold exhibits a remarkably
tunable dependence on the type of gold facet, which we explain using
electronic density functional theory (DFT). This facet dependence
leads to selective surface masking. Next, we 'paint' the iodide-masked
NPs by ligand-mediated polymer grafting, in which polymers selec-
tively adhere only to the unmasked gold surface, rendering patches.
With these two steps demonstrated, we explore the parameter space
of NP shape, iodide concentration, ligand concentration and grafting
temperature, achieving the synthesis of more than 20 different types
of patchy NP, 17 of which are at a yield higher than 80% (Fig. 1, Supple-
mentary Figs. 3–15 and Supplementary Table 1). This approach gen-
erates patch patterns not observed previously, such as face patches,
hybrid face–vertex patches, web-like patches and symmetry-broken
patches. These patterns are predicted by our multiscale scaling

theory[31] and MD simulation performed in parallel with our experi-
ments. Our particles are highly uniform, allowing self-assembly into
millimetre-scale, non-closely packed lattices directed by patches,
which until now have been merely a computational blueprint[32]. The
openness of these lattices can lead to emergent properties, such
as negative Poisson ratios[7] and complete photonic bandgaps[11]. Our
implementation of atomic stencilling is straightforward, based on
the fundamental understanding of interatomic and intermolecular
interactions, and can generalize to other NP systems and substrates
for functional patterning.

## Library of patchy NPs from atomic stencilling
Figure 1a shows an example of traditional stencilling at the macro-
scale, in which a Mona Lisa silhouette-shaped mask is applied onto an
arbitrarily curved pottery surface and painted over. The waxy masked
area repels aqueous ink, delivering intricate patterns on painting.

Likewise, stencilling on NP surfaces is achieved by two steps (Fig. 1b and Supplementary Note 2). During masking, the pristine NPs, which are synthesized following previous literature[33,34], are incubated with sodium iodide. Halide ions are known to adsorb on metal surfaces and are widely used as shape modifiers in NP synthesis[35,36] but have not been used for molecular patterning. Of the three halides (that is, chloride, bromide and iodide), iodide binds most strongly to gold surfaces[35,37] and resists removal from the gold surfaces[35,37], even on extensive washing, which provides stable masking. The incubation with iodide is carefully controlled in degassed water with sodium hydroxides and NPs are subsequently washed, as excess iodide can cause unwanted etching of gold NPs[35] (Methods and Supplementary Note 2). Next, the iodide-masked NPs are mixed with short hydrophobic thiol ligands of 2-NAT and polystyrene-b-polyacrylic acid (PS-b-PAA) in dimethylformamide (DMF) and water, heated at a fixed temperature for 2 h to complete the polymer grafting. 2-NAT covers the unmasked regions and makes them hydrophobic, allowing the physisorption of the PS blocks[24] of the polymers to paint the exposed gold surface. Although other halide ions are present during iodide masking, they show negligible effects on polymer patching (Supplementary Fig. 16a,b). Control experiments using bromide and chloride as masking agents generate non-patchy NPs (Methods and Supplementary Fig. 16c–e).

As summarized in Fig. 1c, the stencil strategy robustly produces a library of patchy NPs for five different core NP shapes (octahedron, bipyramid, cuboctahedron, cube and rhombic dodecahedron), each of which has small vertex truncations. These truncations produce secondary facets (hereafter referred to as vertices), whose atomic arrangement differs from that of the primary facets (hereafter referred to as faces). Varying the reaction conditions allows control over the size, shape and pattern of the patches (Supplementary Figs. 3–15, Supplementary Tables 2–10 and Supplementary Notes 2 and 3). In the absence of iodide masks, NPs are fully coated with polymers regardless of NP shape (Fig. 1c, the 'w/o iodide' region). With iodide masks, moving left to right and top to bottom in Fig. 1c, we see that, for the octahedron and bipyramid shapes, not only is the vertex patch size tunable but the pattern transitions from all-vertex patches to a single vertex patch as the iodide concentration [I⁻] increases while keeping the 2-NAT concentration [2-NAT] constant, breaking the inherent NP symmetry. Cuboctahedra exhibit a transition from all-face to all-vertex patches with increasing [I⁻]/[2-NAT], whereas cubes and rhombic dodecahedra are both dominated by face patches but exhibit further complexity. Their extended face patches can merge into web-like patterns. For rhombic dodecahedra, hybrid patches that include both face and vertex patches can coexist on a NP. Each of the experimentally realized patchy NPs matches that predicted by scaling theory (Fig. 1c). We emphasize that our stencil approach and explicit use of adatom masks differentiates our method from other reports to create polymer patches on NPs[18–24]. The stencilling method is scalable (Supplementary Fig. 17 and Supplementary Note 2.9) and applies to other thiols and block copolymers (Supplementary Fig. 18).

## Facet-dependent iodide masking

Atomic-scale DFT calculations (Methods, Supplementary Tables 11–13 and Supplementary Note 4) show that iodide adsorption onto gold NP surfaces leads to facet-selective binding of 2-NAT ligands, controlled by iodide and ligand concentrations. Experimentally, iodide is expected to oxidize to iodine on metal surfaces[35]; we include this consideration in our DFT model and refer to this process as iodide adsorption. We consider the low-energy (111), (100) and (110) facets of gold observed in the particles[33,34]. Depending on the NP shape, these facets can be large (faces) or small (vertices), an observation that will become important below. Our DFT calculations predict that the Au(111) facet is preferentially masked by iodide at all of the conditions considered (Fig. 2a–c,e, Extended Data Fig. 1 and

Supplementary Figs. 19–23). Figure 2a illustrates the effect of iodide on 2-NAT adsorption at a 2-NAT coverage of 1/3 monolayer (ML) and iodide coverages of 0 and 1/2 ML (top and bottom panels, respectively). In the absence of iodide, 2-NAT resides close to all of the gold surfaces (Fig. 2b) and forms strong chemical bonds (Fig. 2c). As the iodide coverage increases, the Au–S distance $d_{Au–S}$ increases for all surfaces, but the increase in $d_{Au–S}$ is the largest for the (111) surface, followed by the (100) and then the (110) surfaces (Fig. 2b). Concomitantly, the fraction of the binding energy that can be attributed to chemisorption decreases the most for Au(111) and the least for Au(110) as the iodide coverage increases (Fig. 2c). Strong iodide adsorption onto gold NPs is experimentally confirmed by energy-dispersive X-ray spectroscopy (EDX) mapping of iodide-incubated gold NPs (Fig. 2d, Extended Data Figs. 2 and 3 and Supplementary Figs. 24–26). X-ray photoelectron spectroscopy (XPS) and Raman spectroscopy confirm that increasing [I⁻] leads to higher iodide adsorption and less 2-NAT coating onto the NP surface (Extended Data Fig. 4 and Supplementary Tables 9 and 10).

Figure 2e shows a phase diagram delineated into regions based on the equilibrium surface configurations of iodide and 2-NAT adsorption on all three gold surfaces for varying chemical potentials, $\mu_{I^-}$ and $\mu_{2-NAT}$. For 2-NAT, we only include chemisorption configurations. We do observe configurations in which 2-NAT is partially or completely physisorbed (Fig. 2a) but we presume that the weakly bound, physisorbed 2-NAT would not remain on the NP surface to provide stable grafting sites for polymers (Fig. 2c). We use $\Delta\mu_{2-NAT} = \mu_{2-NAT} - E_{2-NAT}^{DFT}$ for plotting Fig. 2e (Supplementary Table 13 and Supplementary Note 4), in which $E_{2-NAT}^{DFT}$ is the energy of 2-NAT in the gas phase and $\Delta\mu_{2-NAT}$ is the rescaled chemical potential of 2-NAT. We see that, with low $\mu_{I^-}$ and high $\Delta\mu_{2-NAT}$, all surfaces are covered by 2-NAT and there is no adsorbed iodide. Conversely, with high $\mu_{I^-}$ and low $\Delta\mu_{2-NAT}$, all surfaces are covered by iodide with no adsorbed 2-NAT. The cyan-coloured intermediate regions in Fig. 2e indicate where stencilling is expected. Each cyan region has 1/3 ML of iodide and no 2-NAT on Au(111) and 1/2 ML 2-NAT but no iodide on Au(110). Our calculations predict competition between iodide and 2-NAT adsorption on Au(100): at low $\mu_{I^-}$, the (100) surface contains 2-NAT only. The surface coverage switches to iodide only as $\mu_{I^-}$ increases with moderate $\Delta\mu_{2-NAT}$ and both iodide and 2-NAT adsorb when the chemical potentials of both species are high.

The DFT calculations predict facet-dependent 2-NAT adsorption on gold in the presence of iodide, which rationalizes the experimental library of patchy NPs. The surfaces of the five gold NP shapes comprise combinations of {111}, {100} and {110} facets (Fig. 3a,b and Extended Data Figs. 5–7). Consistent with DFT calculation, {111} facets are masked first. When all NP faces are {111} and masked by iodide, vertex polymer patches dominate. As shown in Fig. 3c–e, gold octahedra of eight triangular {111} faces have a high yield (92.8%) of six vertex patches (Fig. 3d), with highly uniform patch thickness, position and coverage (Supplementary Figs. 3, 4 and 27 and Supplementary Video 1). Similarly, bipyramids with ten {111} faces exhibit vertex patches on the longitudinal vertices and a doughnut-like patch around the equator, at a yield of 80.4% (Fig. 3f–h, Supplementary Figs. 5 and 6, Supplementary Video 1). When [I⁻] is fixed, variations in [2-NAT] change the patch size (Supplementary Fig. 4a,b). In the control experiment without iodide masking, decreasing [2-NAT] can cause insufficient 2-NAT coverage, leading to partial polymer-coated domains that are irregular and lack facet selectivity (Supplementary Fig. 4c,d).

When NP faces are not all {111}, non-conventional patterns emerge, such as face, web-like and hybrid patches, which are the unique result of facet-selective iodide masking and cannot be created using previously reported strategies[18–24]. For example, cuboctahedra—which have eight faces of {111} and six faces of {100}—can have six face patches on the {100} facets at low [I⁻]/[2-NAT], with {111} facets masked first (Fig. 3i–k, Extended Data Fig. 6f,h, Supplementary Figs. 7 and 8 and Supplementary Video 2). The patch pattern transitions to 12 vertex patches

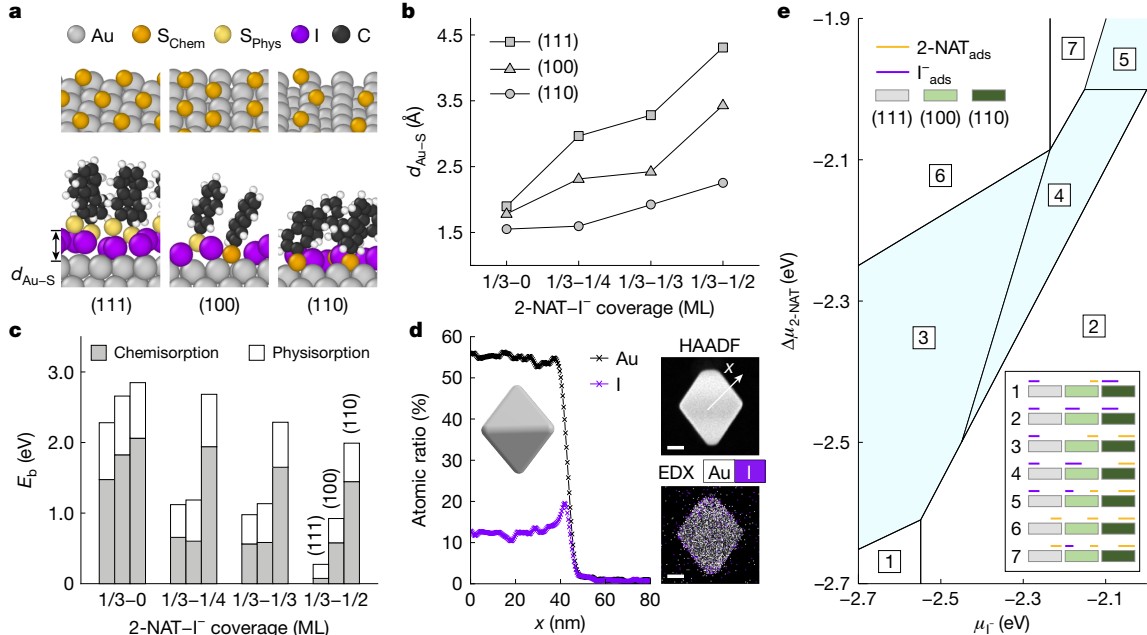

**Fig. 2 | DFT prediction of facet and concentration-dependent stencilling.**
**a**, Geometry-optimized structures in DFT calculations of 2-NAT adsorption without (top) and in the presence of (bottom) iodide on three low-index gold facets of (111), (100) and (110) at surface coverages of 1/2 ML for iodide and 1/3 ML for 2-NAT. Aromatic rings of 2-NAT are omitted for clarity in the top panel. 2-NAT molecules chemisorbed and physisorbed on gold are coloured orange and yellow, respectively. **b**, DFT calculation of $d_{Au-S}$ with increasing iodide coverage on each facet, at a fixed 2-NAT coverage of 1/3 ML. **c**, Contributions to 2-NAT binding energy ($E_b$) to each gold facet from chemisorption (grey bars) and physisorption (white bars) at a fixed 2-NAT coverage of 1/3 ML and increasing iodide coverages from left to right. **d**, The atomic ratio of Au and I in averaged line profiles across the iodide-incubated gold octahedron obtained from EDX (left). The gold octahedron is extensively washed after incubation to remove excess iodide from the solution. High-angle annular dark-field (HAADF) imaging and EDX mapping of the same gold octahedron (right). Scale bars, 20 nm. **e**, Phase diagram delineating equilibrium adsorption configurations on gold as a function of iodide and 2-NAT chemical potentials. The surface coverages of iodide (purple) and 2-NAT (orange) on three different facets are represented by relative line lengths (coverage values in Supplementary Table 13), corresponding to the numbered phase regions in the phase diagram. Regions in which stencil effect is predicted by DFT are shaded in cyan.

when both {100} and {111} faces are masked at high [I⁻]/[2-NAT] (Fig. 3l–n, Extended Data Fig. 6g,i, Supplementary Fig. 9 and Supplementary Video 2). In optimized conditions, the face-patched and vertex-patched cuboctahedra are highly uniform, at yields of 79.4% and 75.4%, respectively (Fig. 3j,m and Supplementary Table 5). Cubes, which contain six {100} faces and eight {111} vertices, exhibit expanded face patches that merge at the cube edges to form web-like patches (Fig. 3o and Supplementary Table 6) at low [I⁻]/[2-NAT]. The patches shrink their size to disconnected face patches at a high yield of 98.0% at high [I⁻]/[2-NAT] (Fig. 3p, Supplementary Figs. 10–12, Supplementary Table 6 and Supplementary Video 1). These observations align with our DFT calculations and Raman results that higher [I⁻]/[2-NAT] leads to decreased 2-NAT adsorption (Extended Data Fig. 4). Rhombic dodecahedron NPs have all 12 faces as {110} facets and two sets of vertex types—six {100} vertices and eight {111} vertices (Fig. 3b and Extended Data Fig. 7a–g). At low [I⁻]/[2-NAT], only the {111} surfaces are masked, leading to a hybrid patch pattern: extended patches on the {110} faces merging into web-like patches with tunable widths, together with small patches, each of diameter 9.0 ± 1.0 nm, precisely located on the six {100} vertices (Fig. 3q, Supplementary Figs. 13a,b,e and 28 and Supplementary Table 8). At high [I⁻]/[2-NAT], the six {100} vertices are also masked, consistent with DFT calculations, leading to only face patches, as all vertices become masked (Fig. 3r and Supplementary Fig. 13c–e). The emergence of these new patch patterns underscores the precise and robust realization of facet-specific masking with nanometre resolution (Supplementary Fig. 29). The stencil approach works for small NPs as well. Smaller rhombic dodecahedra, with 28.6 nm edge length, for example, exhibit the same patch patterns as the larger ones (Extended Data Figs. 7h–k and 8, Supplementary Fig. 14 and Supplementary Video 3).

## Multiscale modelling of patching behaviour

Although iodide masking dictates the available gold NP facets for polymer grafting and, thus, patch patterning, the PS-*b*-PAA block copolymers as patches experience microphase separation and have sizes similar in magnitude to the size of the core NPs. As such, we expect that enthalpic and entropic effects associated with the polymers further fine-tune, within the set of available grafting sites, the patch patterns on NPs and the size of patches[24,31]. To explain these effects, we develop a multiscale modelling workflow to bridge the atomistic-level DFT model with mesoscale polymer behaviour. We first construct a statistical mechanical adsorption model for iodide masking that inputs the binding energies computed by DFT, which determines the stoichiometric ratios of surface binding sites available for polymer grafting on each facet type (Methods, Supplementary Fig. 30 and Supplementary Note 5). In this way, we directly map DFT calculations of individual facets to the 3D NP shapes, thereby quantifying how surface masking affects the thermodynamics that govern chain conformations (Fig. 4a). Microscopically, neighbouring grafted polymers induce crowding, causing chain extension and a large entropic penalty[24]. Meanwhile, chain extension facilitates enthalpically favourable PS–PS contacts (Supplementary Note 6). This competition between conformational entropy and interchain attraction $E_{chain}$ determines the free energy of a grafted polymer chain at the unmasked NP surface locations.

Using this multiscale modelling approach, we predict the patching behaviours as a function of experimentally relevant parameters, including NP shape, NP local surface curvature $\Omega$, [I⁻], grafting concentration and temperature $T$ (Supplementary Notes 6 and 7 and Supplementary Figs. 31–36). In the absence of iodide and excess [2-NAT],

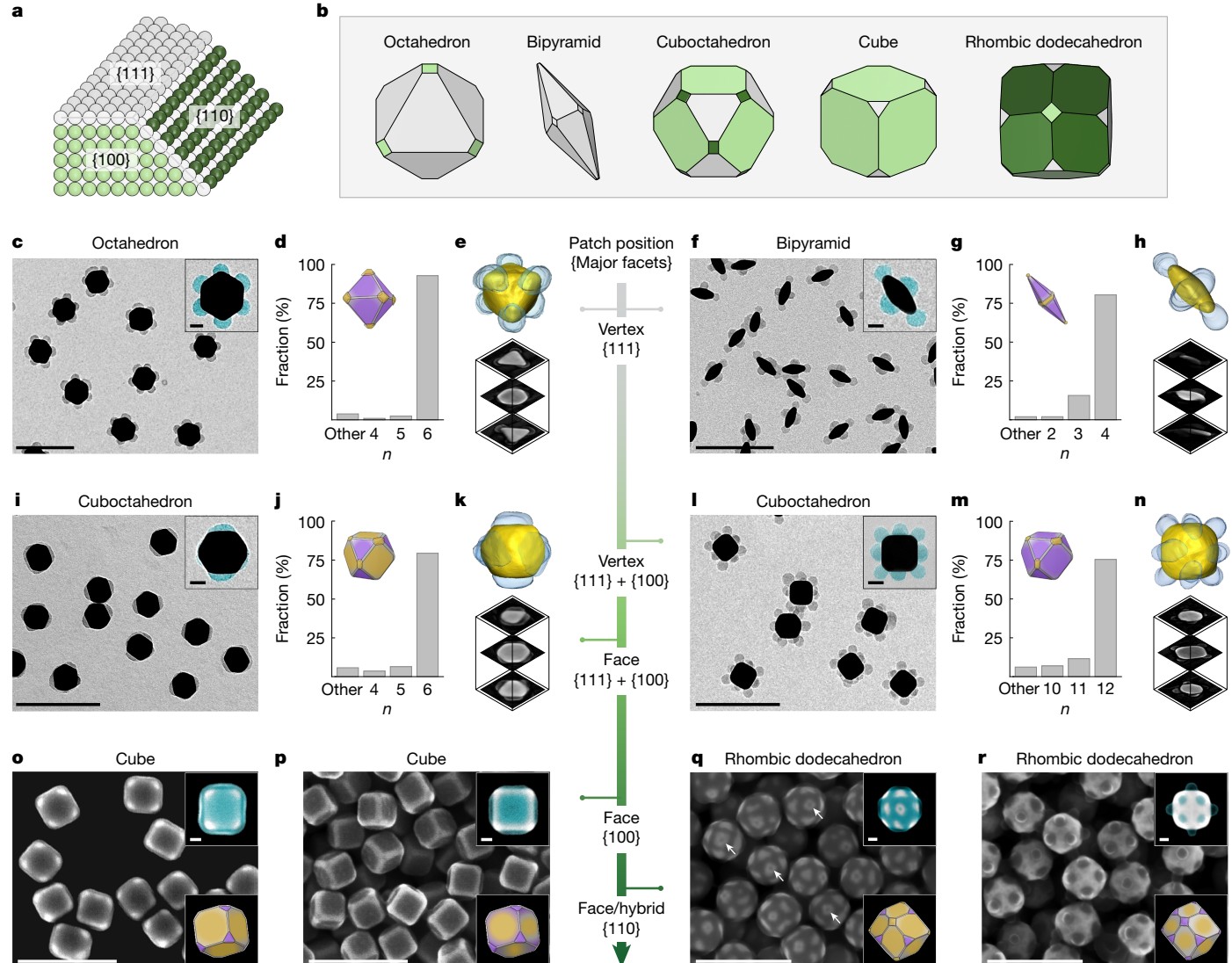

**Fig. 3 | Experimental synthesis of patchy NPs following facet and concentration-dependent stencilling. a,b**, Schematics illustrating low-index facets on gold (**a**) and the correspondingly coloured facets of the five core NP shapes (**b**). **c–e**, Representative TEM image (inset, TEM image with false-coloured patches in cyan) (**c**), histogram of the number of patches on each NP, $n$ (inset, schematic showing iodide (purple) and 2-NAT (orange) covered regions) (**d**) and 3D reconstruction of TEM tomography of the patchy NP and its representative cross-sectional views (**e**). 'Other' in the histogram (**d**) refers to other patch patterns, including fully coated ones. **f–n**, Similar characterizations for patchy bipyramids (**f–h**), face-patched cuboctahedra (**i–k**) and vertex-patched cuboctahedra (**l–n**). **o,p**, SEM images of web-like patched cubes (**o**) and face-patched cubes (**p**). Top inset, SEM image with false-coloured patches in cyan. Bottom inset, schematic showing iodide (purple) and 2-NAT (orange) dominant regions. **q,r**, SEM images of hybrid patched rhombic dodecahedra (**q**) and face-patched rhombic dodecahedra (**r**). Small vertex patches are located on the {100} truncated vertices of hybrid patched rhombic dodecahedra (**q**), some of which are indicated with white arrows. Note that, in the SEM images (**o–r**), the dark regions on the NPs are the polymer patches, as also highlighted in cyan in the insets. More than 100 NPs (208 NPs (**d**), 102 NPs (**g**), 141 NPs (**j**) and 130 NPs (**m**)) are analysed to determine the synthesis yield of each patchy NP type. Scale bars, 200 nm (20 nm for inset images).

our model predicts that NPs are fully coated by polymers (Fig. 4b,f and Supplementary Fig. 31), consistent with our experiments. Following introduction of iodide stencils, grafted chains can no longer freely explore the NP surface but are confined to regions bounded by the masks. In the example of octahedra, vertex patches are found to form in our model (Fig. 4b), MD simulation (Fig. 4c and Supplementary Note 8) and experiments (Fig. 4d and Supplementary Figs. 3 and 4). Figure 4e shows the theoretical prediction of the patch dimensions, including maximum patch thickness $t_m$ and patch coverage fraction $f_{cov}$ (Supplementary Figs. 32–34 and Supplementary Note 3). The initial increase of $t_m$ is associated with increasing [I⁻], reducing the area of available sites as reflected in smaller $f_{cov}$, and thus crowding the chains to induce polymer extension. Similarly, rhombic dodecahedra, although face-patched, exhibit the same dependence of patch dimensions

on [I⁻] as vertex-patched octahedra (Extended Data Figs. 8 and 9, Supplementary Fig. 35, Supplementary Note 8 and Supplementary Video 4). By comparison, cuboctahedra NPs undergo a patch pattern transition (Fig. 3i–n), which is also captured by the model and MD simulation (Fig. 4f–h and Supplementary Video 4).

The patching process is driven largely by thermodynamic equilibrium, although with kinetic formation of intermediates. For example, the patches in octahedra are initially small and only form at a subset of the available vertices (Supplementary Fig. 37), because free polymer chains in solution may not have sampled all vertices. Yet, as more chains adsorb, chain crowding at the vertices drives the patches to the equilibrium state, with more uniform patch size and distribution.

Our predictive multiscale framework is extended across all experimentally relevant handles to construct a phase diagram (Fig. 4i),

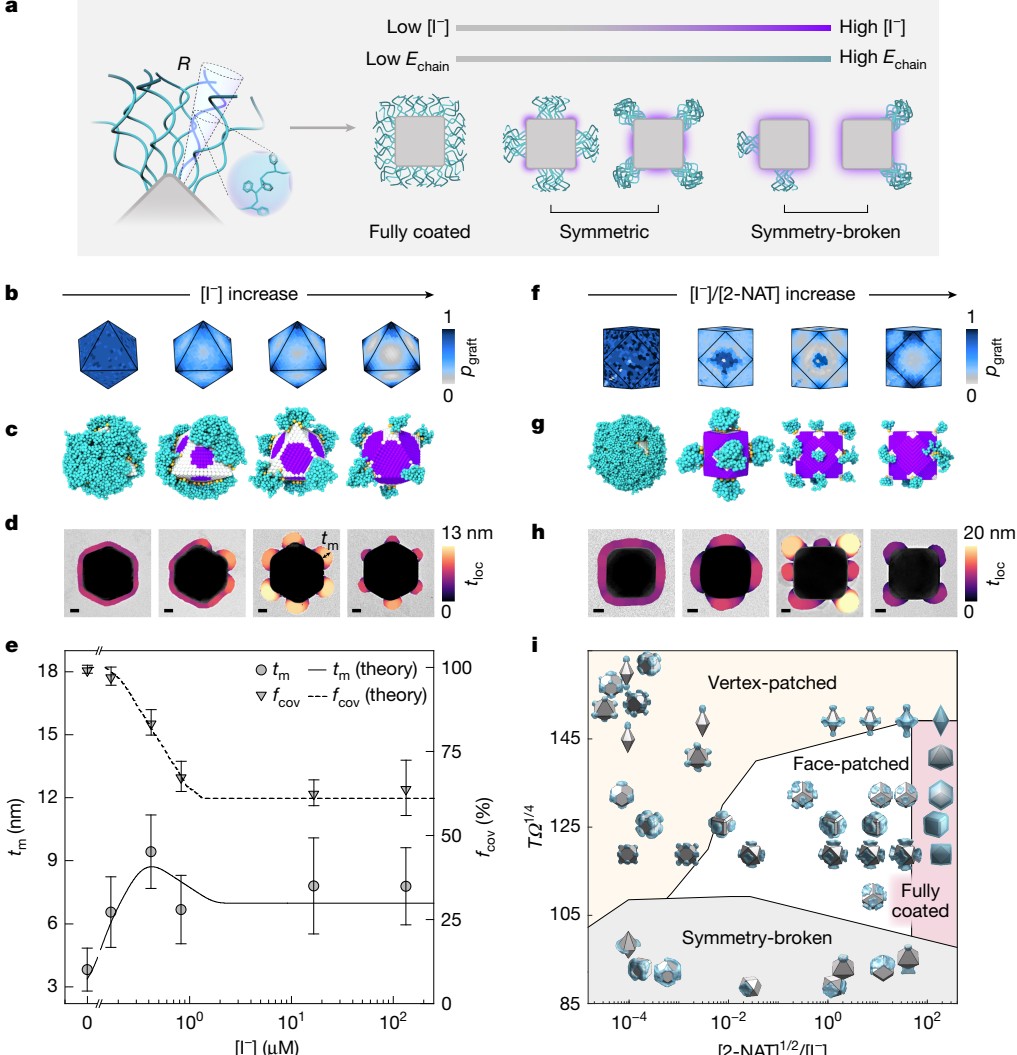

**Fig. 4 | Multiscale theory and MD simulation consistent with experimental observations, demonstrating the tunability and predictability of patch patterns and patch sizes by stencilling. a**, Schematic of the theoretical model of polymer grafting on NP surface and its dependence on [I⁻] and $E_{chain}$. Polymers of end-to-end distance $R$ are grafted with one end with their conformation as a function of $\Omega$, a shape parameter that defines the spatially dependent NP surface curvature (Supplementary Note 6). **b,f**, Polymer grafting probability $p_{graft}$ mapped onto the surfaces of an octahedron (**b**) and a cuboctahedron (**f**). **c,g**, MD simulations (purple, surface regions masked by iodide; cyan, grafted polymer chains) of patchy octahedron (**c**) and patchy cuboctahedron (**g**). **d,h**, TEM images of patchy octahedron (**d**) and patchy cuboctahedron (**h**), with patches colour-coded to their local thickness $t_{loc}$ (Supplementary Note 3).

When two patches overlap in projection, we show the thickness map for one of them for clarity. Reaction conditions from left to right: (**d**) [I⁻] of 0, 0.17, 0.42 and 117.60 µM at a fixed [2-NAT] of 11.3 µM; (**h**) [I⁻]/[2-NAT] of 0, 0.037, 0.980 and 30.7 (Supplementary Tables 2 and 5). Scale bars, 10 nm. **e**, Maximum patch thickness $t_m$ (defined and labelled in **d**) and patch coverage fraction $f_{cov}$ (defined as the fraction of NP surface covered by patches) from theory (lines) and experiment (mean values as symbols, standard deviation as error bars). From low to high [I⁻], a total of 111, 45, 74, 98, 73 and 43 NPs are analysed for each sample, respectively. **i**, Phase diagram of patch patterns predicted by theory. Patchy NP configurations overlaid on the phase diagram are individual theoretical calculations. Coloured regions are approximated by examining the predicted patch patterns.

including four distinct regions of patch patterns: vertex-patched, face-patched, fully coated and symmetry-broken. We use the reduced coordinates of [2-NAT]$^{1/2}$/[I⁻] to reflect the idea that iodide coverage competes with [2-NAT] to control chain coverage on the NP surface. The thermodynamic effects of the polymer chain conformation grafted out of the available sites determined by unmasked regions are captured in reduced coordinate $T\Omega^{1/4}$. A high $T\Omega^{1/4}$ reflects the limit at which steric repulsion between neighbouring chains dominates patch patterning, whereas a low $T\Omega^{1/4}$ indicates that chain–chain attraction dictates the resultant polymer distribution. Notably, the symmetry-breaking limit corresponds to substantial microphase separation between the PS and PAA blocks, in which interchain attraction dominates to break the intrinsic symmetry of iodide-masked NPs (Supplementary Fig. 15).

## Superlattice assembly from patchy NPs

Atomic stencilling provides highly uniform patchy NPs, which allows us to demonstrate their self-assembly into highly ordered superlattices. The polymer patches on neighbouring NPs are repulsive as a result of electrostatic and steric forces arising from the choice of polymer. Consequently, we expect patchy facets to avoid alignment with one another at close range. At the same time, we do not expect the monolayer of iodide on the masked facets to impede the gold–gold van der Waals interactions, so the masked facets are still expected to favour alignment.

Controlled evaporation of the patchy NP solution leads to the formation of superlattices with large interparticle spacing (Fig. 5a–g, Supplementary Figs. 38–47 and Supplementary Table 14). For the

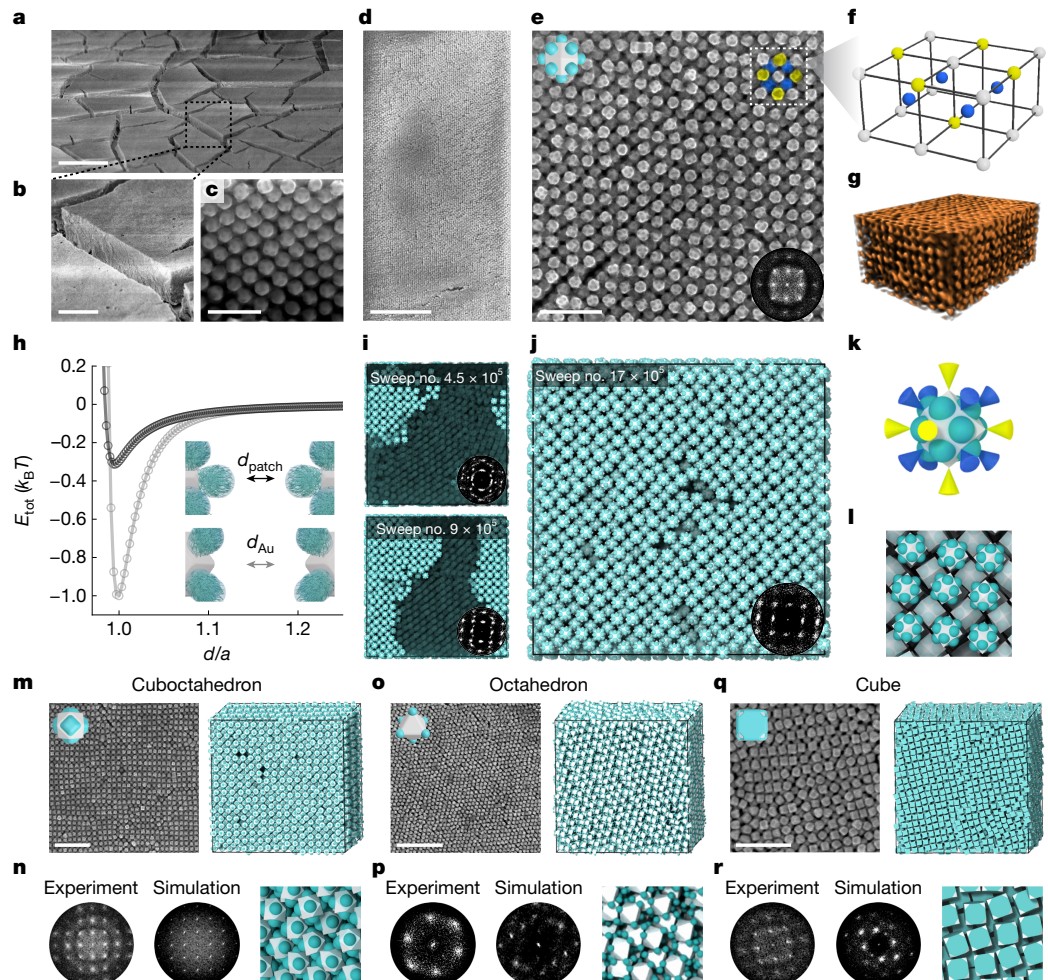

**Fig. 5 | Large-scale self-assembly of patchy NPs into open superlattices.**
**a**–**e**, SEM images of large-scale assembly of face-patched rhombic dodeca-hedra on a glass tube (**a**,**b**) and silicon wafer (**c**–**e**), taken at tilted (**a**–**c**) and top (**d**,**e**) views (Methods). Inset in **e**, schematic of NP and fast Fourier transform (FFT) of the assembled lattice. **f**, 3D schematics of the lattice motif in **e**, with the nearest neighbours (blue) and next-nearest neighbours (yellow). **g**, X-ray tomography reconstruction of the lattice. **h**, Potential of mean force calculations of the pairwise interactions of two face-patched rhombic dodecahedra in two different configurations (inset schematic) as a function of the interparticle distance $d$ normalized by NP circumsphere diameter $a$. **i**,**j**, MC simulation snapshots of patchy rhombic dodecahedron assembly. Insets, FFT of the lattices. **k**, Directional interaction (conical shape) of the particle with the neighbours coloured the same as in **e** and **f**. **l**, Zoom-in view of **j**, with top-layer NPs highlighted for visualization. **m**,**o**,**q**, SEM images (left) and MD simulation (right) of face-patched cuboctahedra (**m**), vertex-patched octahedra (**o**) and face-patched cubes (**q**). Inset in **m**,**o**,**q**, NP schematic. **n**,**p**,**r**, Corresponding FFT from experiment (left) and MC simulation (middle) and zoomed-in view of simulation (right). Scale bars, 20 μm (**a**); 5 μm (**b**); 200 nm (**c**); 2 μm (**d**); 300 nm (**e**); 500 nm (**m**,**o**,**q**).

four patchy NPs as building blocks (face-patched rhombic dodeca-hedra, face-patched cuboctahedra, vertex-patched octahedra and face-patched cubes), their superlattices feature single-crystalline domains that extend over hundreds of micrometres or even millimetres were it not for the cracks that develop during drying. These superlat-tices exhibit body-centred cubic (BCC) or body-centred tetragonal (BCT) symmetries, which differ from the closely packed lattices formed by pristine NPs of the same shapes[27] (Supplementary Fig. 48 and Sup-plementary Table 15). Our Monte Carlo (MC) simulations predict these superlattices and reveal that the directional interactions among pre-cisely patterned patches are key to obtaining these open structures (Fig. 5h–r, Supplementary Fig. 49, Supplementary Note 9 and Sup-plementary Video 5).

Consider face-patched rhombic dodecahedra as an example. They have 12 patched faces and 14 masked vertices, so we propose that each particle will favour having 14 particles in its first and second neigh-bour shells with the attractive vertices aligned (Fig. 5k). A top-down view of the superlattice shows an interparticle spacing of 77.5 ± 6.9 nm (1.56 times the NP size) in one two-dimensional (2D) square-symmetry lattice layer (Fig. 5e). 3D reconstruction from X-ray tomography at

single-particle resolution confirms a BCC superlattice, exceeding tens of layers in the $z$-dimension (Fig. 5g and Supplementary Figs. 50–52). Successful formation of the same superlattice on different substrates demonstrates that interparticle interactions, rather than substrate effects, dominate the assembly process (Supplementary Figs. 39–42). The BCC superlattice has a volume fraction as low as 0.31, in sharp contrast to the space-filling face-centred cubic superlattices assem-bled from pristine rhombic dodecahedra (Supplementary Fig. 48). We compute potentials of mean force for pairs of face-patched rhombic dodecahedra at different relative orientations (Fig. 5h). MC simula-tions using these interaction potentials reveal that initially randomly oriented patchy NPs reorient to produce BCC lattices with long-range ordering (Fig. 5i,j and Supplementary Fig. 53). As suggested, the BCC lattice is stabilized by the formation of a unit cell featuring a central patchy NP surrounded by eight nearest neighbours and six next nearest neighbours, in both simulation and experimental assemblies (Fig. 5f,k,l and Supplementary Fig. 54).

All four patchy NPs form into large-scale superlattices, with agree-ment between experiment and MC simulation. The face-patched cuboctahedra (Fig. 5m,n and Supplementary Figs. 43–46 and 55) and

vertex-patched octahedra (Fig. 5o,p and Supplementary Figs. 47 and 56) assemble into BCC structures at low volume fractions. The face-patched cubes assemble at a low volume fraction (0.33 in MC simulation; Supplementary Figs. 57 and 58) into a BCC superlattice, which transitions to a BCT superlattice (lattice vector ratio of 1.8:1:1) at increased volume fraction (0.53 in MC simulation in Supplementary Fig. 57, experiment in Fig. 5q,r), in which a BCC arrangement is geometrically impossible. Notably, the patch pattern, size and thickness are all crucial for the formation of long-range, open structures. The assembly of symmetry-broken patchy NPs shows disrupted local order. Furthermore, if the patches are too small, the strong attraction from gold–gold van der Waals forces between exposed faces can lead to kinetically trapped aggregates (Supplementary Fig. 59).

## Conclusion

Our work presents bottom-up stencilling of NPs using adatom adsorption to create masks. Strong interatomic bonding leads to nanometre precision in molecular patterning, resulting in a large library of patchy NPs. We develop a robust multiscale theoretical framework for atomic stencilling, by bridging DFT calculations, polymer scaling theory, MD simulation of chain distributions and configurations with MC simulation of large-scale assembly, which predicts and confirms our experimental observations. This experiment–theory–simulation integration can extend to other NP systems, in which the tunability of core NP composition, shape, size, as well as patch polymer chemistry is limitless. For example, gold nanorods are promising candidates worth further study owing to their rich faceting behaviours determined by particle size and synthesis conditions[38]. Stencilling can also be applied to other metal NPs such as palladium nanocubes (Supplementary Fig. 60), in which the presence of iodide leads to the formation of face-patched palladium nanocubes, similar to the face-patched gold nanocubes. Iodide adsorption on Pd(111) and Pd(110) has been studied[39,40], although the specific 2-NAT co-adsorption mechanism needs further investigation. The general principle of facet-selective adsorption and masking is likely to generalize to other metals, such as Cu (refs. 36,41,42) and Ag (ref. 43). Surface chemistry is critical to the self-assembly[27,28], electron–photon coupling[44], charge/electron transfer[45] and chemical reactivities[10] of NPs and other surfaces. Using atomic stencilling to control surface chemistry is poised to accelerate NP applications, from metamaterials to quantum information systems, fuel cells, batteries and catalysis[5,6,8–10], as well as to achieve nanometre-scale patterning on substrates[46] for integrated circuits and other multifunctional materials.

Our work demonstrates nanoscale colloidal valency control using explicit surface ligand patches for 3D superlattice engineering. 'Valency' control has been shown with extensive successes in simulation[32], at the molecular scale (for example, metal–organic frameworks) and for micrometre-sized colloids[16], leading to innovations in reticular chemistry, tandem catalysis and understanding of out-of-equilibrium dynamics owing to symmetry-breaking. At the nanoscale, valency control has been demonstrated using NPs uniformly grafted with DNA[47]; here we show that directional interactions can be rationally achieved from surface patchiness, which may or may not reflect the underlying particle symmetry. Experimental observation of the self-assembly dynamics of patchy NPs, using liquid-phase TEM[48–50] or small-angle X-ray scattering, can potentially reveal how the directional interactions govern the assembly pathways.

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

## Methods

### Chemicals

All chemicals were used without further purification after purchase: gold(III) chloride trihydrate (≥99.9% trace metals basis, $HAuCl_4 \cdot 3H_2O$, Sigma-Aldrich), sodium borohydride (99%, $NaBH_4$, Sigma-Aldrich), sodium iodide (99.999%, NaI, Sigma-Aldrich), sodium bromide (≥99.99%, NaBr, Sigma-Aldrich), sodium chloride (99.3%, NaCl, Fisher Scientific), potassium iodide (≥99.5%, KI, Sigma-Aldrich), silver nitrate (≥99.0%, $AgNO_3$, Sigma-Aldrich), sodium tetrachloropalladate(II) (approximately 36%, $Na_2PdCl_4$, Acros Organics), L-ascorbic acid (Bio-Xtra, ≥99.0%, AA, Sigma-Aldrich), sodium hydroxide (99.99%, NaOH, Sigma-Aldrich), cetyltrimethylammonium chloride (>95%, CTAC, TCI, product number: H0082), benzyldimethylhexadecylammonium chloride (≥97.0%, BDAC, Sigma-Aldrich), cetylpyridinium chloride (>98.0%, CPC, TCI, product number: H0078), sodium citrate tribasic dihydrate (≥99.0, $C_6H_5Na_3O_7$, Sigma-Aldrich), $PS_{154}$-$b$-$PAA_{51}$ ($M_n = 16,000$ for the PS block and $M_n = 3,700$ for the PAA block, $M_w/M_n = 1.04$, Polymer Source Inc.), $PS_{154}$-$b$-poly(acrylamide)$_{49}$ ($M_n = 16,000$ for the PS block and $M_n = 3,500$ for the poly(acrylamide) block, $M_w/M_n = 1.15$, Polymer Source Inc.), $PS_{154}$-$b$-poly(ethylene oxide)$_{170}$ ($M_n = 16,000$ for the PS block and $M_n = 7,500$ for the poly(acrylamide) block, $M_w/M_n = 1.09$, Polymer Source Inc.), 2-NAT (99%, Sigma-Aldrich), biphenyl-4-thiol (97%, Sigma-Aldrich), DMF (anhydrous, 99.8%, Sigma-Aldrich) and hydrochloric acid (99.999% metals basis, HCl, Alfa Aesar). Nanopure deionized (DI) water (18.2 MΩ cm at 25 °C) purified by a Milli-Q Advantage A10 system was used. Note that the purity of cetyltrimethylammonium bromide (CTAB) is important for gold NP synthesis. We have used three types of CTAB: (1) Sigma-Aldrich, for molecular biology, ≥99%, product number: H6269; (2) Sigma-Aldrich, BioXtra, ≥99%, product number: H9151; and (3) Sigma-Aldrich, BioUltra, ≥99.0%, product number: 52365. For the data presented in this work, CTAB (1) and CTAB (2) were used for the experiments performed before February 2024. CTAB (3) has been used to reproduce the experiments afterwards.

### Synthesis of core NPs

Gold octahedron, cuboctahedron, rhombic dodecahedron, cube and bipyramid NPs were synthesized following previously reported methods with slight modifications[33,34] (Supplementary Note 1). Palladium cubes were synthesized following previous literature[51] with modifications (Supplementary Note 2.8).

### Synthesis of patchy NPs

Patchy NPs were synthesized in two steps, which are iodide masking and ligand-mediated polymer grafting. Using the patchy octahedra in Fig. 3c as an example, for the iodide-masking step, we first degas DI water by purging with $N_2$ for 1 h to minimize oxygen content. This degassed water is used throughout the masking step to minimize the oxidative etching of NPs. A gold octahedra solution was centrifuged at 2,400 × $g$ for 15 min and the resulting pellet was diluted in 20 mM CTAC with degassed water to achieve a final NP concentration of 0.5 optical density (OD) at a maximum extinction wavelength $\lambda_{max}$ (Stock Solution I). Next, 5.75 µl of freshly prepared iodide solution (1 ml of 10 mM NaI, 500 µl of 200 mM NaOH and 8.5 ml degassed DI water) was added dropwise to 6.9 ml of Stock Solution I under mild vortex and incubated undisturbed for 30 min. After the incubation, 2.1 ml of 40 mM CTAB was added, followed by three rounds of centrifugation (3,400 × $g$, 5,300 × $g$ and 2,200 × $g$ for 15 min each). The supernatant was removed after each centrifugation and a 10-µl pellet was redispersed in 10 ml of 20 mM CTAB after the first round and 1 ml of DI water after the second round, respectively. The final pellet from the third centrifugation, after supernatant removal, was diluted with DI water to reach 5.0 OD at $\lambda_{max}$ with 0.07 mM CTAB (Stock Solution II). The final CTAB concentration can be varied depending on differently shaped NPs, as provided in Supplementary Note 2.

For ligand-mediated polymer grafting, still using the patchy octahedra in Fig. 3c as an example, we sequentially added 817 µl DMF, 5 µl of 2-NAT solution (2 mg ml$^{-1}$ in DMF), 200 µl of Stock Solution II and 80 µl of PS-$b$-PAA (8 mg ml$^{-1}$ in DMF) into an 8-ml vial under mild vortex. The vial was tightly capped with a Teflon-lined cap, sonicated for 5 s, Parafilm-sealed and heated at 110 °C in an oil bath for 2 h without disturbance. After cooling to room temperature in the oil bath, the reaction mixture was transferred to a 1.5-ml microcentrifuge tube and centrifuged three times at 3,400 × $g$, 1,300 × $g$ and 1,000 × $g$ (15 min each) to remove molecular residues. After each centrifugation, 1.49 ml of the supernatant was removed and a 10-µl pellet was redispersed in 1.49 ml of water. Following the final centrifugation, the 10-µl pellet was diluted with 90 µl of water for subsequent characterizations and self-assembly experiments. Detailed descriptions of the experimental procedure of the synthesis of other patchy NPs are provided in Supplementary Note 2 and Supplementary Tables 2–8.

Similar procedures with modifications can be extended to palladium nanocubes (Supplementary Note 2.8), other thiols and block copolymers (Supplementary Notes 2.1 and 2.5) and scale-up synthesis (Supplementary Note 2.9).

### Large-scale self-assembly of patchy NPs

For self-assembly experiments, a patchy NP solution in a 1.5-ml microcentrifuge tube was left undisturbed overnight, allowing the particles to sediment and concentrate (≥30.0 OD at $\lambda_{max}$).

**Coffee ring effect-driven assembly on a silicon wafer.** A silicon wafer (3 × 3 mm², Ted Pella) was cleaned with water, acetone and isopropanol through sonication for 5 min each, followed by 60 s of $O_2$ plasma treatment using a Harrick Plasma PDC-32 (maximum RF power of 18 W) to make the surface hydrophilic. Meanwhile, a 4 × 4-inch² piece of Parafilm was placed on a TechniCloth (TX609, Texwipe) with a water-filled Petri dish cover (60 × 15 mm², Falcon) on top. 3.5 µl of a concentrated patchy NP solution was drop-casted onto the wafer on the Parafilm. Immediately after drop-casting, both the silicon wafer and Petri dish were covered by a large Petri dish (100 × 15 mm², VWR) and gently pressed down to seal against the Parafilm, maintaining a humid environment (humidity: 75–80%). A typical drying process takes 16–24 h at room temperature (Supplementary Fig. 38a).

**Capillary force-driven assembly in a glass capillary tube.** A 10-µl glass capillary tube (inner diameter: 0.5573 mm, Drummond Scientific Company) was filled with 2 M NaOH, incubated for 20 min, rinsed with water and dried to make the inner wall hydrophilic before use. 5 µl of concentrated patchy NP solution was then drawn into the capillary tube, keeping the NP solution several centimetres away from both tube ends. The capillary tube was suspended inside a loosely capped 15-ml centrifuge tube, which was then placed in a desiccator until the solution had fully dried, forming a visible gold ring around the interior. See the experimental set-up in Supplementary Fig. 38b.

### Electron microscopy characterizations

**TEM, SEM and STEM characterizations.** TEM images were acquired with a JEOL 2100 LaB6 transmission electron microscope operated at 200 kV. SEM images were captured using an FEI Helios NanoLab 600i operated at 2 kV with a beam current of 0.17 nA and a working distance of 4.0 mm. HAADF-scanning transmission electron microscopy (STEM) imaging was conducted on a probe aberration-corrected Thermo Fisher Scientific Themis Z scanning transmission electron microscope operated at 300 kV. STEM-EDX mapping was acquired with a Thermo Fisher Scientific 'Kraken' Spectra 300 operated at 60 kV equipped with Dual-X EDX detectors with a collection angle of 1.76 sr.

**Sample preparation for HAADF-STEM characterization.** For NP facet imaging and analysis, solutions of pristine NPs were drop-casted on

TEM grids, with CTAB concentration reduced through three rounds of centrifugation. Approximately 50–200 μl of the pristine NP solutions in 40 mM CTAB was used per sample. Each sample was first diluted with 1 ml of water in 1.5-ml microcentrifuge tubes, followed by removing 990 μl of supernatant after each of the first two centrifugations. After each centrifugation, a 10-μl pellet was redispersed in 990 μl of water. Following the third centrifugation, a 1.5-μl pellet was drop-casted onto a carbon-coated copper TEM grid (Electron Microscopy Sciences, CF400-Cu) and dried completely. Before imaging, samples were plasma-cleaned for 2 min at 15 W under a 12-sccm flow of Ar + $O_2$ to remove residual CTAB molecules covering NPs using a PIE Scientific Tergeo-EM plasma cleaner.

**HAADF-STEM imaging.** HAADF images were acquired with a beam current of approximately 20 pA and semi-convergence angle of 18 mrad. The camera length was 115 mm and the collection angles of the HAADF detector were 62–200 mrad for image acquisition. To minimize image distortion from sample drift, several sequential frames (10–50 frames) with short dwell time (100–500 ns) were acquired, which were used to render drift-corrected frames to enhance the contrast.

**TEM tomography.** TEM images for tomography reconstruction were collected by titling the samples from 0° to −60° and then from 0° to +60° in 2° intervals, capturing 61 images per sample. To minimize beam damage, a low electron dose rate of 6–8 e⁻ Å⁻² s⁻¹ was used. To improve the polymer patch contrast, a defocus of −2,048 nm was used. Image alignment and contrast-transfer function correction were performed using IMOD64 4.9.3 software[52]. Tomograms were generated using OpenMBIR with diffuseness of 0.3 and smoothness of 0.2 as reconstruction parameters, which uses a model-based iterative algorithm of tomogram reconstruction[53]. 3D models of the tomograms were visualized using Amira 6.4 from Thermo Fisher Scientific. The tomograms were denoised using three filters: median (3 × 3 × 3 voxel neighbourhood, iterations 26), Gaussian filter (kernel size 9, 3 × 3 × 3 standard deviation) and edge-preserving smoothing (time 25, step 5, contrast 3.5 and sigma 3). Polymer patches and gold NPs were segmented by greyscale intensity thresholding and refined manually in Amira 6.4 software.

**STEM-EDX characterization.** Samples for the STEM-EDX analysis were prepared by drop-casting Stock Solution II on carbon-coated copper TEM grids (Electron Microscopy Sciences, CF400-Cu). The octahedra and cuboctahedra were iodide-masked. To minimize carbon contamination, the TEM grids were baked overnight at 130 °C in high vacuum to remove excess CTAB. STEM-EDX maps were acquired with a 120-pA probe current, a 30-mrad semi-convergence angle and continuous raster-scanning with drift correction, using a 2-μs dwell time over approximately 4 h. For iodide-masked cuboctahedra, the NPs were predominantly oriented on the carbon support along the [001] direction. Therefore, the stage was tilted 45° to the [110] zone axis, aligning both the {111} and {100} facets 'edge-on'. Elemental mapping was achieved by fitting and quantifying the peak intensities above the background using the Cliff–Lorimer method[54]. Two high-quality EDX maps of seven cuboctahedra were selected and a total of 1,692 line profiles were obtained (Supplementary Fig. 25). Similarly, iodide-masked octahedra were oriented on the carbon support along the [110] direction and their 'edge-on' {111} facets were used for iodide concentration analysis (Extended Data Fig. 2).

## TEM image analysis

**Patch local thickness $t_{loc}$.** The $t_{loc}$ of patches was determined as the diameter of the largest circle that can fit within the patch and includes the target pixel. Patch contours were manually outlined in ImageJ to define the patch regions. The $t_{loc}$ map was then obtained using the built-in 'Local Thickness' function in ImageJ, as also detailed in our previous work[24].

**Maximum patch thickness $t_m$ and patch coverage fraction $f_{cov}$.** The values of $t_m$ and $f_{cov}$ were measured using a neural-network-based method on TEM images[55]. The neural network-based TEM image segmentation is detailed in Supplementary Note 3 and Supplementary Figs. 32 and 33. After training the neural network with manually labelled TEM images of diverse patchy NP shapes, a shape fingerprint $t$ was extracted from the patch contours. First, the centroid of the gold NP was identified and rays extending from $\theta = -180°$ to $\theta = 179°$ at 1° intervals were drawn from the centroid. The distance that each ray travels within the patch region was recorded as a function of $\theta$. $t_m$ is determined by the maximum value of $t$ and $f_{cov}$ is identified as the range of $\theta$ in which $t$ is non-zero. See detailed description in Supplementary Note 3 and Supplementary Figs. 34–36.

## Other characterizations

**UV–Vis measurements.** Ultraviolet–visible (UV–Vis) spectra were measured using a Scinco S-4100 PDA spectrophotometer with a quartz cuvette (path length = 1 cm, VWR).

**XPS characterization.** XPS analysis was performed using a Kratos Axis Ultra equipped with a monochromatic Al Kα radiation X-ray source and an energy resolution of 0.4 eV. Before characterization, 10 μl of each Stock Solution II, prepared with various iodide concentrations, was drop-casted on a silicon wafer cleaned with water, acetone and isopropanol and then fully dried. The silicon wafers were fixed to the sample bar and transported to the instrument in an airtight container under an Ar atmosphere for XPS signal acquisition. Data processing and peak fitting were conducted using the CasaXPS software. Atomic concentrations within samples were calculated by integrating the fitted XPS spectra for all analysed regions and adjusting for atomic relative sensitivity factors. See detailed sample conditions in Supplementary Table 9.

**Raman characterization.** Raman characterization was performed using a Horiba XploRA-nano TERS/TEPL with a 100× objective lens. The Raman samples were prepared following two steps, without polymer grafting (Supplementary Table 10). First, gold octahedra were iodide-masked in varying [I⁻] and then concentrated to reach 20.0 OD at $\lambda_{max}$ in 0.07 mM CTAB, following the iodide-masking procedure described above. Next, 40 μl of 2-NAT (0.2 mg ml⁻¹ in DMF) and 200 μl of the iodide-masked octahedra were sequentially added into 860 μl of DMF in a vial with mild vortex. The remaining reaction condition and washing steps follow the standard iodide masking procedure, except for centrifugation, which was performed three times at 2,900 × g for 15 min and 2,400 × g for 7 min twice. After the first two centrifugations, the pellet was resuspended in 1 ml of DMF. After the final centrifugation, the sediment was redispersed in 20 μl of DMF and the NP concentration was adjusted to 13.0 OD at $\lambda_{max}$ by adding water. A 5-μl aliquot of this NP solution was drop-casted onto a clean glass slide. The glass slide was used after washing with isopropyl alcohol and water and then fully dried. Raman measurements were performed with a 638-nm excitation wavelength, 1-mW laser power and three scans for average (90 s exposure time each).

**X-ray tomography of large-scale self-assembly.** The X-ray tomography sample was prepared through a focused-ion beam milling using the FEI Helios NanoLab 600i. Before milling, platinum was deposited on the self-assembled patchy rhombic dodecahedra on a silicon wafer, with a deposition diameter of 3 μm and a thickness of 300 nm. Subsequently, the assembly was milled into a cylinder with a diameter of 2.5 μm. The shaped cylinder was lifted with an OmniProbe and mounted onto a tungsten needle tip (Supplementary Fig. 50).

X-ray tomography data were acquired at the hard X-ray nanoprobe beamline of National Synchrotron Light Source II at Brookhaven National Laboratory. A monochromatic beam at 12 keV was selected

and then focused by a set of crossed multilayer Laue lenses to produce a nanobeam[56] with a size of approximately 13 nm. 2D fly-scans were performed in a grid pattern with at least 150 × 150 pixels, 50-ms dwell time and 10-nm step size at every 2° to collect 91 projections covering 180°. Far-field diffraction patterns were analysed with a ptychographic reconstruction algorithm to retrieve both the complex-valued probe and the object functions. The acquired fluorescence spectra were fitted using the software package PyXRF. Individual 2D frames were coarsely aligned using ImageJ 1.5, MultiStackReg plugin, whereas fine alignments were adjusted with a cross-correlation function in Tomviz 1.9, followed by 3D reconstruction. The data were further processed by Fourier filtering using the sharp lattice peaks in the reciprocal space image of the superlattice to remove noise and sharpen the particle positions for subsequent segmentation, using Dragonfly 2020.2 software[57].

### DFT calculations of gold surfaces with iodide and 2-NAT

All DFT calculations were performed using the Vienna Ab initio Simulation Package[58-60] with projector augmented waves[61]. The generalized gradient approximation by Perdew, Burke and Ernzerhof was used for the exchange-correlation functional[62]. We chose an energy cut-off of 450 eV as an optimal value for our plane-wave basis set. For the sampling of the first Brillouin zone, Monkhorst–Pack grids were used[63]. We also included the DFT-D3 method of Grimme et al. with the Becke–Johnson damping to describe long-range van der Waals interactions[64]. See details in Supplementary Note 4.

### Computation and theory modelling of patchy NPs and their assemblies

**Patchy NP grafting simulation.** A library of individual patchy NPs was simulated using MD using the HOOMD-blue simulation engine[65]. First, a single anisotropic particle was placed at the centre of the simulation box. Polymer chains with one grafting end were then randomly distributed on the surface of the central anisotropic particle and allowed to freely move on the surface to find their equilibrium positions. The location of iodide-masked regions predicted from our iodide adsorption theory (see details in Supplementary Note 5) were modelled as spherical beads that are strongly repulsive to the polymer chains. All chain–chain interactions exhibit Lennard–Jones attraction with each other to capture the PS–PS aggregation in experimental conditions. All other interactions were purely repulsive and modelled using the Weeks–Chandler–Andersen repulsive potential. For detailed description, see Supplementary Note 8.

**Theoretical prediction of patches.** Polymer patch patterns and sizes were theoretically predicted using MC grafting simulation involving two steps: (1) determining the surface distribution of iodide-masked region and (2) placing polymer chains onto the unmasked ('free') surface sites, while accounting for chain–chain interactions. We began by constructing a grid of points on the core NP surface, defining potential sites for iodide or polymer attachment. Using the Metropolis algorithm with surface energies computed from DFT, iodide-masked points were placed on the various surface sites (Supplementary Note 5). Placing iodide-masked points on the particle surface uses the Metropolis algorithm, guided by surface energies computed from DFT. Iodide-masked points cannot accommodate polymer grafting, leaving the remaining open locations as the only 'free' sites for polymer grafting. After an occupancy matrix logs iodide-masked positions, the first polymer chain grafts on the surface based on its Boltzmann-weighted free energy as a function of surface locations. The matrix updates with this chain attachment and 'free' surface sites within a correlation length $\xi$ of any polymer-occupied sites gain a favourable chain–chain interaction, governed by the Flory–Huggins parameter $\chi$ (Supplementary Note 6). This entire process repeats for each polymer chain until the target grafting density is achieved. See Supplementary Note 7 for details.

**Simulation of patchy NP self-assembly.** MC simulations were performed to obtain the self-assemblies of patchy NPs, using the HOOMD-blue simulation engine[65]. We first defined a patchy particle whose patch locations and sizes are commensurate with those measured from the above MD simulation and validated with theory and experiments. Patches were modelled to exhibit hard sphere interactions with each other to capture the strong PAA–PAA electrostatic repulsions between the outer surface of the patches (Supplementary Fig. 49). The core NP interactions were modelled using a Kern–Frenkel attraction between the various surface sites that were not covered by polymeric patches. See Supplementary Note 9 for details.

## Data availability

The data that support the findings of this study are available from the corresponding authors on request. The raw TEM and SEM images of the 21 types of patchy NP listed in Supplementary Table 1 can be found at https://doi.org/10.13012/B2IDB-4862788_V1.

## Code availability

The custom MATLAB codes for patch shape prediction are provided at https://github.com/chenlabUIUC/Atomic_stencil. The custom MATLAB codes for image processing and analysis are available from the corresponding authors on request.

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

**Acknowledgements** The experimental synthesis, self-assembly and main characterizations of patchy NPs were supported by the US Department of Energy (DOE), Office of Science, Office of Basic Energy Sciences, Division of Materials Science and Engineering, under award number DE-SC0020723 (A.K., C.K., Q.C.). These experiments were carried out in part at the Materials Research Laboratory (MRL) Central Research Facilities, University of Illinois. A.K., C.K. and Q.C. appreciate R. T. Haash at MRL for assistance with XPS. The polymer scaling theory and simulations (T.W., T.V., S.C.G.) were supported by the US National Science Foundation (NSF) under Cooperative Agreement No. 2243104, 'Center for Complex Particle Systems (COMPASS)' Science and Technology Center and in part by a Vannevar Bush Faculty Fellowship sponsored by the Department of the Navy, Office of Naval Research under ONR award number N00014-22-1-2821. Computational resources and services are provided by Advanced Research Computing at the University of Michigan. DFT calculation (E.M.K., Junseok Kim, K.A.F.) used Bridges-2 at the Pittsburgh Supercomputing Center through allocation DMR110061 from the ACCESS programme, which was supported by NSF grant no. 2138259, no. 2138286, no. 2138307, no. 2137603 and no. 2138296, and by the US DOE, Division of Materials Science and Engineering, under award number DE FG02-07ER46414. The STEM-EDX

studies (Y.-T.S., D.A.M.) were supported by the Department of Defense, Air Force Office of Scientific Research under award FA9550-18-1-0480 and made use of the Cornell Center for Materials Research (CCMR) facilities supported by the NSF MRSEC programme (DMR-1719875), NSF MIP (DMR-2039380), NSF-MRI-1429155 and NSF (DMR-1539918). Y.-T.S. and D.A.M. thank J. Grazul, M. S. Ramos and M. Thomas for technical support and maintenance of the electron microscopy facilities. The X-ray tomography (A.M., Z.G.) work used the Hard X-ray Nanoprobe (HXN 3-ID) beamline of the National Synchrotron Light Source, a US Department of Energy Office of Science User Facility operated for the Department of Energy Office of Science by Brookhaven National Laboratory under contract no. DE-AC02-98CH10886.

**Author contributions** A.K. and Q.C. conceived the idea and designed the experiments. A.K., C.K., J.R.C., L.Y., E.Y., S.M. and Q.C. performed the experiments and data analysis. T.W. and S.C.G. performed MC and MD simulations and analysis. T.V. and S.C.G. developed the polymer theory. E.M.K., Junseok Kim and K.A.F. performed DFT calculations and analysis. Y.-T.S., A.K. and D.A.M. performed STEM-EDX mapping and analysis. F.C.K., C.K., Y.-S.L., H.A. and A.K. performed electron tomography and reconstruction. A.M., C.K. and Z.G. conducted X-ray tomography and reconstruction. Y.Z. performed X-ray scattering of the self-assembled lattices. C.-Y.H., Jiyeon Kim and C.K. performed HAADF-STEM imaging. A.K. and Q.C. wrote the first draft of the main text. C.K. and A.K. wrote the first draft of the Supplementary Information. All authors contributed to the editing of the paper. Q.C., S.C.G. and K.A.F. supervised the work.

**Competing interests** The authors declare no competing interests.

**Additional information**
**Correspondence and requests for materials** should be addressed to Kristen A. Fichthorn, Sharon C. Glotzer or Qian Chen.

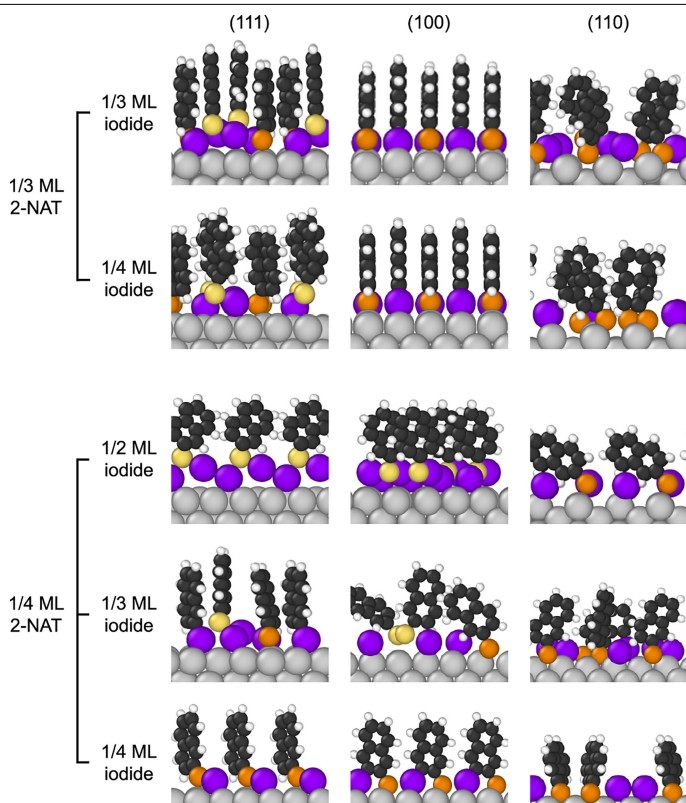

**Extended Date Fig. 1 | Optimized adsorption configuration for various 2-NAT and iodide coverages on gold surfaces predicted by DFT.** Images from DFT calculations of co-adsorbed 2-NAT and iodide on Au(111), Au(100) and Au(110) at various surface coverages (units: ML). Coloured spheres represent atoms: gold (grey), iodine (purple), physisorbed sulfur (yellow), chemisorbed sulfur (orange), carbon (black) and hydrogen (white). The configuration for 1/3 ML–1/2 ML 2-NAT–iodide coverage is shown in Fig. 2a.

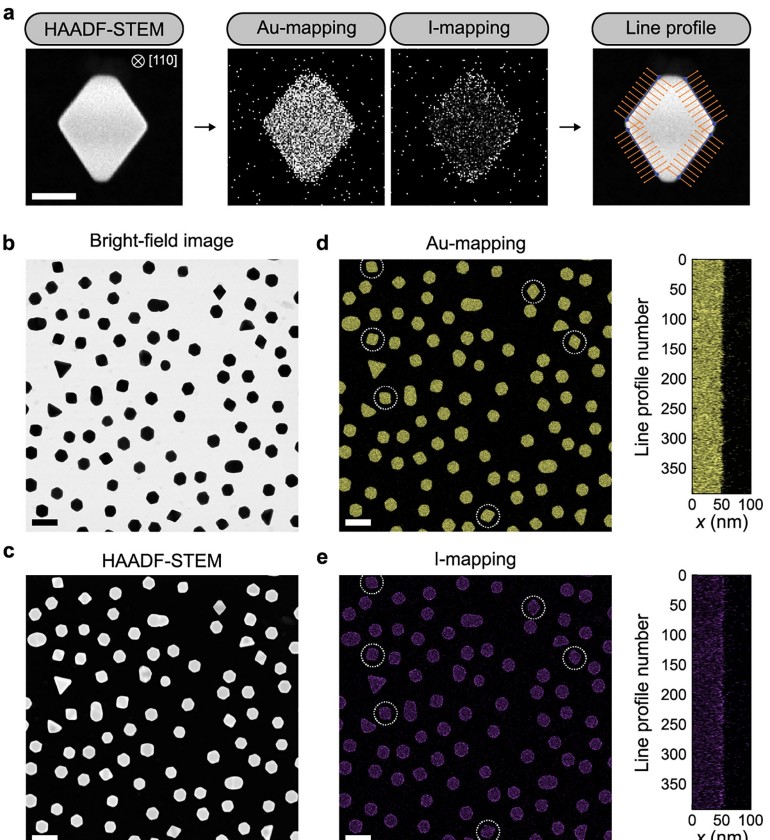

**Extended Data Fig. 2 | STEM-EDX analysis of iodide-masked octahedra.**
**a**, The workflow of STEM-EDX analysis for iodide-masked octahedra. The particle shown in Fig. 2d is used here to illustrate the workflow. **b**–**e**, Low-magnification bright-field image (**b**), HAADF-STEM image (**c**), Au EDX map (**d**) and I EDX map (**e**) with corresponding stacked line profiles, from which atomic ratios of Au and I are obtained. The reduced shape uniformity of gold NPs in this sample is presumably because of the extensive surfactant removal (Methods) required for high-sensitivity STEM-EDX detection of submonolayer iodide, which may destabilize unprotected gold surfaces. For EDX line profile analysis, we focus on intact octahedra oriented in the [110] projection (examples highlighted with dotted circles in **d** and **e**). Stacked line profiles are obtained along the contour of the projected octahedra as illustrated in the line profile step in **a**. A total of 400 line profiles are stacked in **d** and **e**. Scale bars, 50 nm (**a**); 200 nm (**b**–**e**).

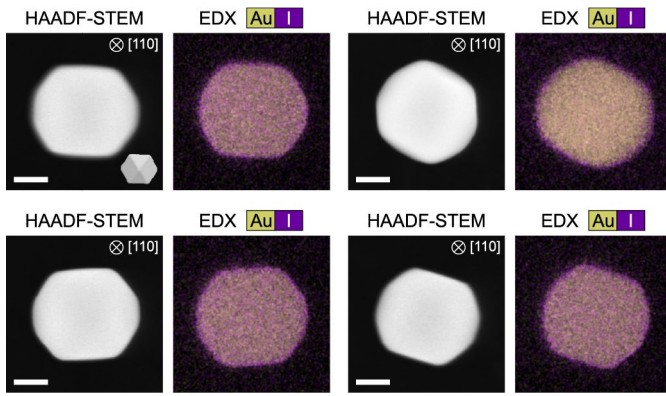

**Extended Data Fig. 3 | STEM-EDX characterization of iodide-masked cuboctahedra.** HAADF-STEM images and EDX maps of Au and I of iodide-masked cuboctahedra, represented in yellow and purple, respectively. Scale bars, 20 nm.

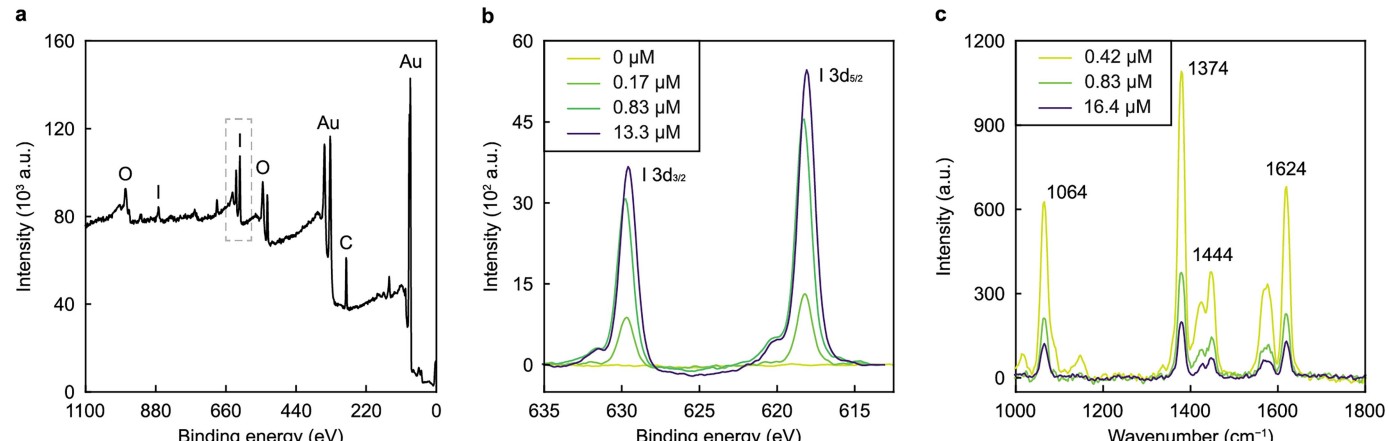

**Extended Data Fig. 4 | XPS and Raman characterizations of iodide-masked octahedra showing [I⁻] effect. a**, XPS of gold octahedra incubated with [I⁻] of 13.3 µM. **b,c**, XPS (**b**) and Raman spectra (**c**) of gold octahedra incubated with varied [I⁻]. As [I⁻] increases from 0 to 13.3 µM, more iodides are adsorbed on the gold NP surfaces, resulting in higher I peak intensities in XPS characterizations presented in **b**. As iodide adsorption on the gold NP surface increases with [I⁻] increase from 0.42 to 16.4 µM, less 2-NAT is bound to the NP surface at a fixed [2-NAT] of 46.0 µM, resulting in decreased 2-NAT peak intensities in Raman spectra presented in **c**. For the detailed sample preparation, see Methods and Supplementary Tables 9 and 10.

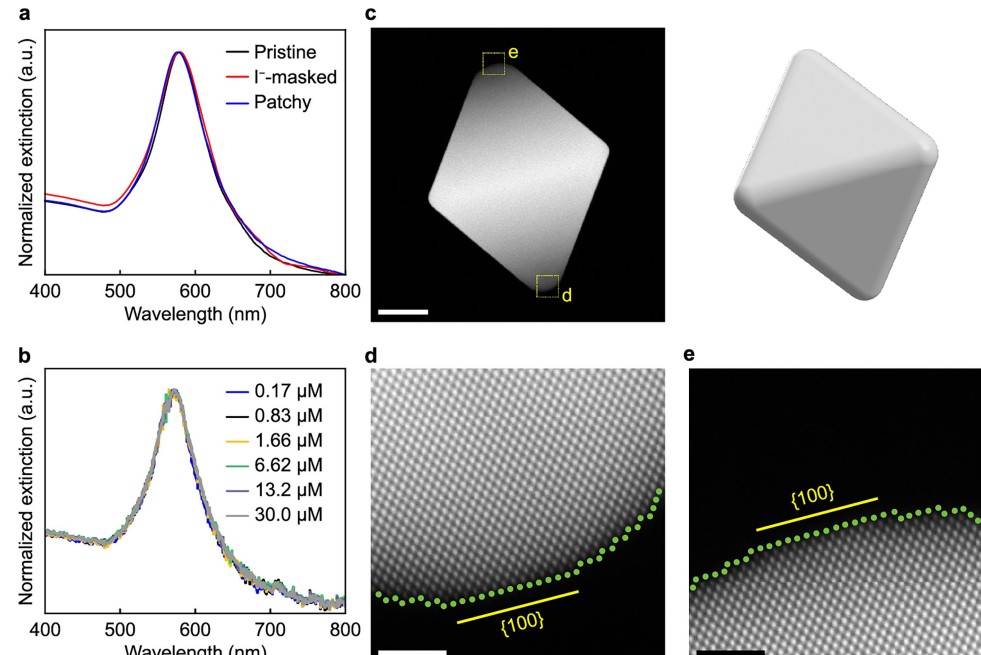

**Extended Data Fig. 5 | UV-Vis spectra and HAADF-STEM images of octahedra.**
**a**, Normalized UV-Vis spectra of the pristine (black), iodide-masked at [I⁻] of
0.25 μM (red) and patchy (blue) octahedra synthesized at [I⁻] of 0.25 μM and
[2-NAT] of 17.0 μM. **b**, Normalized UV-Vis spectra of iodide-masked octahedra at
varying [I⁻] from 0.17 to 30.0 μM. No localized surface plasmon resonance shift is
observed after iodide masking, confirming that the core gold NPs maintain their
ensemble size and shape. **c**, HAADF-STEM images of an unmasked gold
octahedron and the corresponding 3D model. **d**,**e**, Zoomed-in HAADF-STEM
images of the vertex regions shown by boxes in **c**. The vertices exhibit {100}
terraces as denoted by yellow lines. Scale bars, 20 nm (**c**); 2 nm (**d**,**e**).

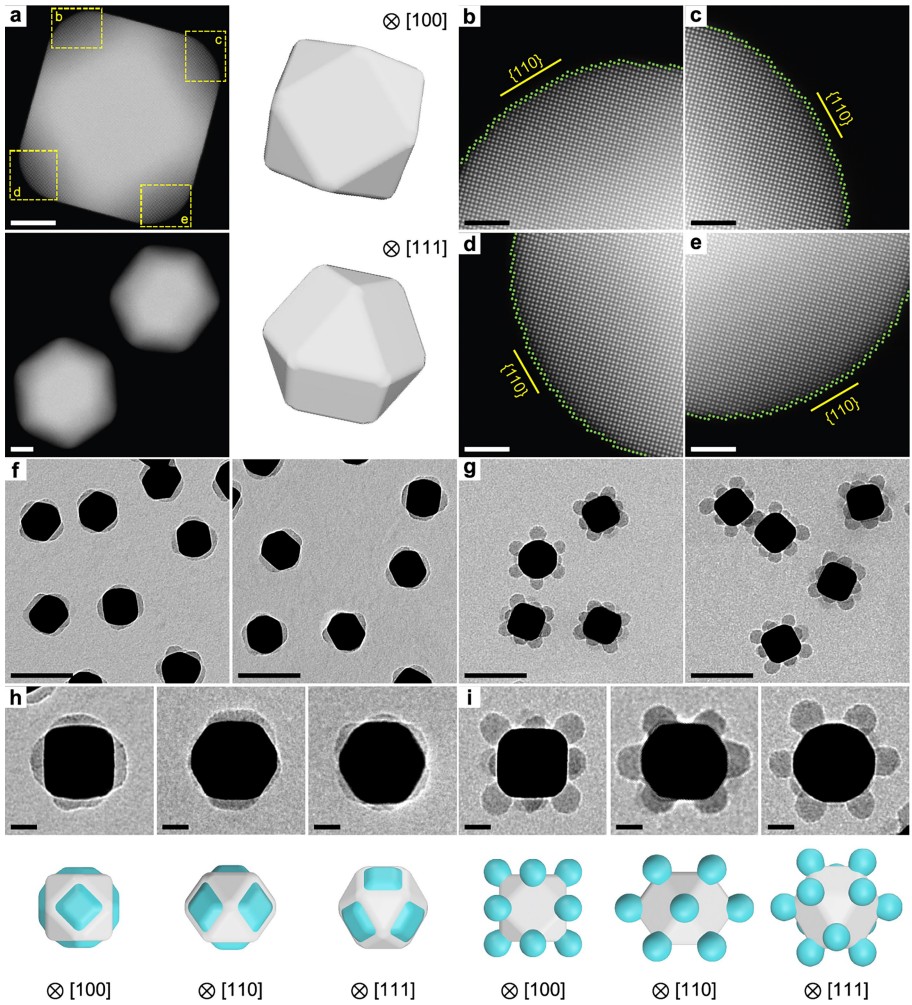

**Extended Data Fig. 6 | Surface facets of gold cuboctahedra and examples of patchy cuboctahedra. a**–**e**, HAADF-STEM images of unmasked gold cuboctahedra and corresponding 3D models. **b**–**e**, Zoomed-in views of the particle vertex regions shown by boxes in **a**. The vertices exhibit {110} terraces as denoted by yellow lines. **f**,**g**, Representative TEM images of face-patched (**f**) and vertex-patched cuboctahedra (**g**). **h**,**i**, Zoomed-in TEM images and corresponding 3D models of face-patched cuboctahedra (**h**) and vertex-patched cuboctahedra (**i**) viewed along different orientations. Scale bars, 10 nm (**a**); 2 nm (**b**–**e**); 100 nm (**f**,**g**); 20 nm (**h**,**i**).

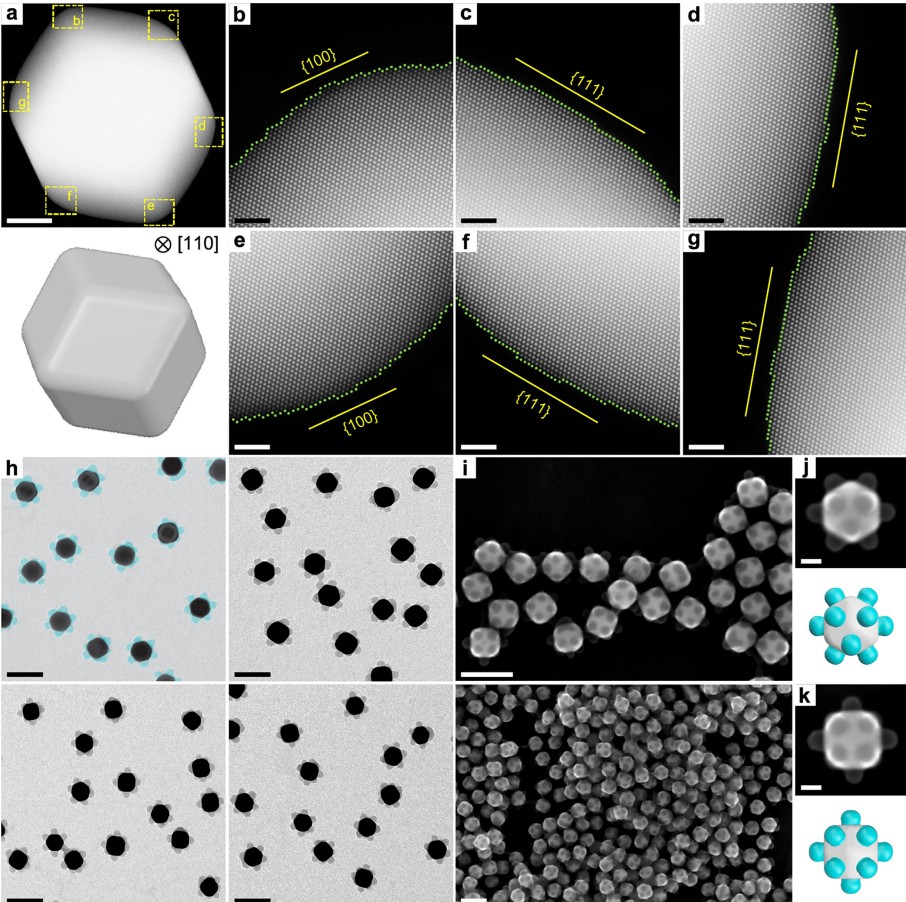

**Extended Data Fig. 7 | Surface facets of gold rhombic dodecahedra and examples of patchy rhombic dodecahedra. a–g**, HAADF-STEM images of an unmasked gold rhombic dodecahedron (Supplementary Fig. 2d) and corresponding 3D model. **b–g**, Zoomed-in views of the vertex regions shown by boxes in **a**. The vertices exhibit {111} and {100} terraces (denoted by yellow lines). **h,i**, Representative TEM (**h**) and SEM (**i**) images of patchy small rhombic dodecahedra (Supplementary Fig. 2c). **j,k**, SEM images and corresponding schematics of patchy small rhombic dodecahedra viewed along the [111] (**j**) and [100] (**k**) directions, respectively. Scale bars, 20 nm (**a**); 2 nm (**b–g**); 100 nm (**h,i**), 20 nm (**j,k**).

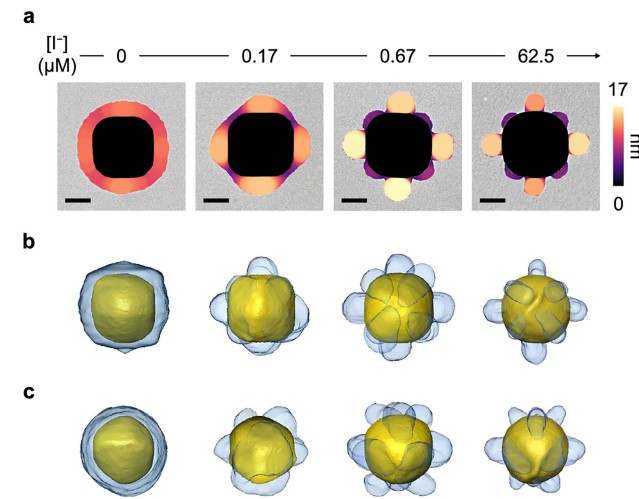

**Extended Data Fig. 8 | Patch pattern transition of patchy rhombic dodecahedra. a**, Representative TEM images of patchy small rhombic dodecahedra overlaid with patches colour-coded to their local thickness $t_{loc}$. Reaction conditions from left to right: [I⁻] of 0, 0.17, 0.67 and 62.5 μM at a fixed [2-NAT] of 56.7 nM. **b**,**c**, Corresponding 3D reconstruction of patchy rhombic dodecahedra from TEM tomography shown along the [100] (**b**) and [111] directions (**c**). Scale bars, 20 nm.

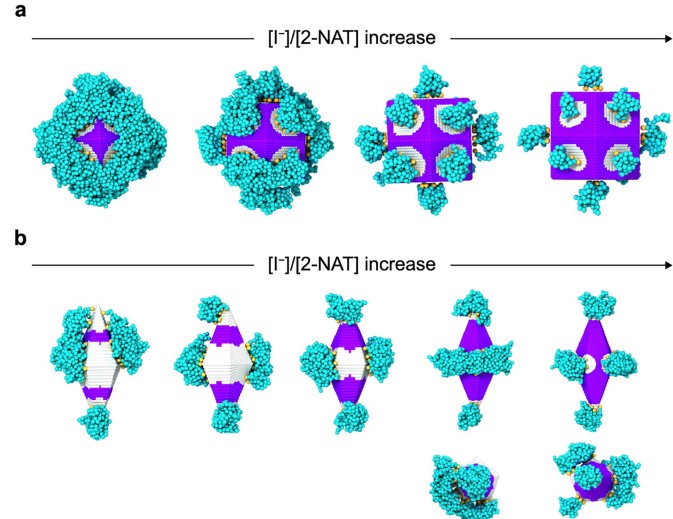

**Extended Data Fig. 9 | MD simulation of the formation of patchy NPs with varying [I⁻]/[2-NAT]. a,b**, MD simulation of patchy rhombic dodecahedra (**a**) and bipyramids (**b**). Increasing [I⁻]/[2-NAT] induces a transition from fully coated NPs to stencilled patchy NPs and symmetry-broken patchy NPs. In **b**, the top and bottom rows present side and top views, respectively.