## [Peer Review File · Nature]

Patchy Nanoparticles by Atomic Stencilling

Corresponding Author: Professor Qian Chen

Version 0:

Reviewer comments:

Referee #1

(Remarks to the Author)

This manuscript presents a significant advancement in the bottom-up fabrication of precisely patterned nanoparticles (NPs) at nanoscale. The authors report a robust method for producing uniform and diverse gold patchy nanoparticles using atomic-scale “stencilling” with halide ions, specifically iodide, followed by ligand-mediated polymer grafting. The “stencilling” process is finely tunable by modulating the structure of iodide submonolayers through simple variation of iodide ion concentration. Transmission electron microscopy (TEM) combined with density functional theory (DFT) calculations indicates that patch formation arises from the selective adsorption of iodide ions onto (111) facets of gold nanoparticles at low iodide concentrations, leaving the unmasked regions available for polymer grafting. This facet-selective masking was convincingly demonstrated across a variety of gold nanoparticle morphologies, including octahedra, cuboctahedra, rhombic dodecahedra, cubes, and bipyramids. Furthermore, these well-defined patchy nanoparticles can self-assemble into superlattices, highlighting their potential as versatile building blocks in nanostructured materials. Overall, this is an elegant and practical approach for creating uniform patchy nanoparticles with promising applications in targeted delivery, catalysis, self-assembly, and nanomaterials engineering. I recommend publication of this manuscript following minor revision. Below are a few questions and suggestions for clarification:

1. The manuscript describes the successful synthesis of many distinct types of patchy nanoparticles and demonstrates that iodide ions preferentially adsorb on the (111) facets of gold nanoparticles under low-concentration conditions. While those results are absolutely fascinating, will the decrease of 2-NAT give similar outcomes? A minor thing: the authors show patch formation on gold nanoprisms at their tips—does this suggest broader applicability to other anisotropic shapes such as nanorods?
2. Will the selective adsorption of iodide ions be unique to gold? Could this “stencilling” strategy be extended to other noble or coinage metals (e.g., silver or copper) with comparable surface chemistries? If so, do iodide ions also preferentially mask the <111> facets of these metals, or is the observed selectivity specific to gold-iodide coordination chemistry?
3. During nanoparticle synthesis, halide ions such as Cl^- and Br^- are commonly introduced. What is their fate post-synthesis, and do they compete with or influence the adsorption behavior of iodide ions? What role do other surface capping ligands play in the adsorption behavior? For instance, where would cetyltrimethylammonium bromide be after the formation of polymer domains.

Referee #2

(Remarks to the Author)

A. Summary of the key results

The authors present a novel bottom-up approach to nanoparticle (NP) surface patterning, termed “atomic stencilling”, which uses iodide adsorption as a molecular mask and ligand-mediated polymer grafting as a “paint” to create patchy nanoparticles. This method enables the synthesis of over 20 distinct patchy NP morphologies with high yield and precision. Authors use a combination of density functional theory (DFT), molecular dynamics (MD), and Monte Carlo (MC) simulations to explain and predict patch formation and NP self-assembly into non-close-packed superlattices. The co-adsorption of iodide and 2-naphthalene thiol on gold depends on the type of gold facet, which they explain using DFT, leading to selective surface masking that allows us to achieve various patch patterns. Then, they “paint” the iodide-masked NPs by ligand-

mediated polymer grafting, in which polymers selectively adhere only to the unmasked gold surfaces, rendering patches.

B. Originality and significance: if not novel, please include reference

This work is original and a novel contribution to the field of nanoparticle surface engineering and self-assembly. The atomic stencilling approach is both conceptually elegant and experimentally robust, offering a new paradigm for creating anisotropic, functional nanomaterials. This work has major implications across nanotechnology, materials science, and applied physics.

Specific parts that I consider original (and cool!):

- The idea of using atomic-scale masking via iodide adsorption is highly original and bridges a gap between top-down lithography and bottom-up self-assembly.
- The ability to control NP surface chemistry at the nanometer scale opens new avenues in colloidal crystal engineering, metamaterials, drug delivery, etc.
- Demonstration of millimeter-scale superlattices with BCC and BCT symmetry is a significant result!

C. Data & methodology: validity of approach, quality of data, quality of presentation; D. Appropriate use of statistics and treatment of uncertainties; E. Conclusions: robustness, validity, reliability

Their approach is a synergistic integration of synthesis, advanced characterization (e.g., STEM-EDX, XPS, Raman, tomography), and multiscale modeling. I don't know how much "polymer theory" helped here, so I would suggest they remove that. But the use of DFT to rationalize facet-selective iodide adsorption and its impact on ligand binding is rigorous and well-supported. Computational prediction of patch morphology and self-assembly behavior also a strength. Phase diagrams and simulations provide a powerful tool for designing future patchy NPs. Conclusions seem robust and justified.

F. Suggested improvements: experiments, data for possible revision

While the method is elegant, the scalability of the iodide masking and polymer grafting process is not addressed. Maybe I missed this but what about the sensitivity to reaction conditions (e.g., iodide concentration, temperature, CTAB levels); could that limit reproducibility outside controlled lab settings.

The study focuses exclusively on gold nanoparticles. Can the authors comment on how extendable this idea of atomic stencilling is to other materials (e.g., silver nanoparticles)?

The patches are primarily polymer-based. How extendable is this to other chemical ligands that may have specific roles for catalytic, magnetic, or fluorescent functions?

Is it fair to expect thermodynamics to be the primary driving force; what about polymer chain dynamics during grafting and kinetics of assembly (e.g., diffusion of particles)? Given the high impact nature of this journal, I think it is fair for me to ask for some additional temporal (experimental) characterization to observe patch formation and at least nanoparticle assembly dynamics (again I hope I didn't miss this - the supplementary information is so large that it is easy to miss a section!); I did note the simulation movies in SI.

G. References: appropriate credit to previous work?

Yes appropriate credit has been given to previous work.

H. Clarity and context: lucidity of abstract/summary, appropriateness of abstract, introduction and conclusions

Overall, the manuscript is well-written, with clear figures and logical flow.

Abstract (the first thing any reader would see) is also written in a way any scientist (not just a materials scientist) can appreciate the work.

Supplementary information provides critical experimental and computational details; a minor suggestion is for the authors to also create concise summaries of the main protocols to improve readability/accessibility.

Referee #3

(Remarks to the Author)

A. Summary of the key results

The authors report a novel way to make patchy nanoparticles of various geometric shapes (octahedron, bipyramid, cuboctahedron, cube, and rhombic dodecahedron) using an atomic, bottom-up approach analogous to macroscopic "stencilling", that involves controlled and selective masking of gold nanoparticle facets with iodide ions followed by polystyrene-*b*-polyacrylic acid polymer brush attachment to unmasked particle areas such as vertex truncations mediated by a short hydrophobic thiol ligand (2-naphthalenethiol, 2-NAT). The authors show TEM, EDX mapping, TEM tomography and SEM images which show superb control of the polymer lobes and their positioning as a function of iodine/2-NAT ratio. Furthermore, the authors show results from DFT, MD, MC grafting and MC simulations that both confirm and predict iodine and 2-NAT binding specificity to various Au surfaces, molecular arrangement of polymer chains, polymer patch patterns, and patchy particle assembly. Finally, patchy particles are dried using either coffee ring assembly or capillary drying and show formation of long-range ordered structures on micro to millimeter length scales.

B. Originality and significance:

The authors present an original approach for the large-scale synthesis of patchy gold nanoparticles, that is driven by a competition of crystal lattice binding energy and entropy contributions and is potentially generalizable to other nanoparticle systems. The approach is different from previous nanoparticle patching approaches as iodine is used as a masking agent.

C. Data & methodology: validity of approach, quality of data, quality of presentation

The manuscript is very well written with a concise and systematic presentation of the various steps in the patching process supporting the validity of the approach, showing a deep understanding of the mechanistic details, and an excellent command of characterization and simulation tools. The data is of superb quality and the aesthetic of the figures is very pleasing.

D. Appropriate use of statistics and treatment of uncertainties

The authors use histograms to show patch distributions and %yield to indicate the quality of their patching process. It would be useful if the authors would define what is meant by “achieving the synthesis of more than 20 different patchy NPs in high yield (Fig1, Supplementary Figs. 3-15)” besides “simply” showing TEM images of a few particles per synthesis. Maybe the yield could be expressed in terms of the amount of gold nanoparticle seeds used and patchy particles made.

E. Conclusions: robustness, validity, reliability

The authors present convincing arguments that their patching approach is robust for gold nanoparticles of various geometries and that patches are formed in reliable fashion. It is not apparent how easy this patching process can be translated to other nanoparticle systems as the authors claim in the conclusion “This experiment-theory-simulation approach can immediately be extended to other NP systems,…” Gold is quite special in its chemical properties and binding interactions with other materials, which is likely the reason why it was chosen by the authors in the first place.

F. Suggested improvements: experiments, data for possible revision

A few revisions are recommended as described below, but no additional data or experiments are needed.

G. References: appropriate credit to previous work?

The authors have cited relevant literature and given credit to previous work.

H. Clarity and context: lucidity of abstract/summary, appropriateness of abstract, introduction and conclusions

The authors have provided a well-written manuscript that is densely packed with a large amount of insightful information and delightful results.

Comments to improve the manuscript:

Page 3 Line 102, “With these two important steps demonstrated, we explore…”

Please, define “high yield” in this sentence. Figure 1 and the supporting info only supply images of a few particles that show patch uniformity, but do not provide information about the yield.

Figure 1. It seems that w/o iodine row should come after the w/iodine row to follow the sequence in the manuscript text.

Page 6 Lines 200-201, “We do observe configurations in which 2-NAT is partially…”

Please, explain why it is presumed that the physisorbed 2-NAT will be washed off.

Figure 2e Readability of the $\Delta\mu$ 2-NAT vs μ l- plot could be improved. Maybe it would be easier to mark the smaller fields with numbers and then put a list of the numbers and the corresponding three-color bar images in the big white space on the bottom right or maybe connect the color bars with lines to the fields they refer to.

Figure 4 Caption. Please, define Ω also in the caption.

Page 11 lines 349-353, “Usage of T…”

This sentence is not clear and could be improved.

Page 14, Conclusion, lines 433-436

Gold seems to be rather unique in its ability to support reversible binding and not react with surface ligands or adatoms. It would be helpful if the authors could provide at least one other example system where they predict a similar approach could work. If such a system is not easy to identify, maybe the authors could at least provide guidance on what characteristics such a system would have to exhibit.

Extended Data Fig 2.

The variability of the gold nanoparticle shapes in this figure is very surprising and requires at least a comment as to why this data is shown. The synthesis seems to involve nanorods followed by etching processes leading to very uniform gold nanoparticle seeds as shown in the supporting info, but ED Figure 2 shows the often-encountered variability in gold nanoparticle shapes.

General comment:

It is somewhat surprising that the assembly experiments are only shown for the most symmetrically modified patches. Although this may be outside the scope of this already very dense paper, curiosity drives the following questions: Have the authors tried assembly of the non-symmetrically patched nanoparticles? Were difficulties encountered in these systems? Or do the calculations predict interesting structures for asymmetric patch geometries?

Version 1:

Reviewer comments:

Referee #1

(Remarks to the Author)

The authors have adequately addressed my comments. I now fully endorse the publication of the manuscript as it is.

Referee #2

(Remarks to the Author)

I think the authors have satisfactorily addressed the reviewer comments. I recommend this interesting publication.

Referee #3

(Remarks to the Author)

The authors have done an excellent job in addressing the various questions and comments from the three reviewers. The manuscript has been improved further and is ready for publication.

Manuscript ID: 2025-03-08125

Title: Patchy Nanoparticles by Atomic “Stencilling”

Authors: Ahyoung Kim, Chansong Kim, Tommy Waltmann, Thi Vo, Eun Mi Kim, Junseok Kim, Yu-Tsun Shao, Aaron Michelson, John R. Crockett, Falon C. Kalutantirige, Eric Yang, Lehan Yao, Chu-Yun Hwang, Yugang Zhang, Yu-Shen Liu, Hyosung An, Zirui Gao, David A. Muller, Kristen A. Fichthorn, Sharon C. Glotzer, Qian Chen

A. Summary of the key results

The authors report a novel way to make patchy nanoparticles of various geometric shapes (octahedron, bipyramid, cuboctahedron, cube, and rhombic dodecahedron) using an atomic, bottom-up approach analogous to macroscopic “stencilling”, that involves controlled and selective masking of gold nanoparticle facets with iodide ions followed by polystyrene-*b*-polyacrylic acid polymer brush attachment to unmasked particle areas such as vertex truncations mediated by a short hydrophobic thiol ligand (2-natphthalenethiol, 2-NAT). The authors show TEM, EDX mapping, TEM tomography and SEM images which show superb control of the polymer lobes and their positioning as a function of Iodine/2-NAT ratio. Furthermore, the authors show results from DFT, MD, MC grafting and MC simulations that both confirm and predict iodine and 2-NAT binding specificity to various Au surfaces, molecular arrangement of polymer chains, polymer patch patterns, and patchy particle assembly. Finally, patchy particles are dried using either coffee ring assembly or capillary drying and show formation of long-range ordered structures on micro- to millimeter length scales.

B. Originality and significance:

The authors present an original approach for the large-scale synthesis of patchy gold nanoparticles, that is driven by a competition of crystal lattice binding energy and entropy contributions and is potentially generalizable to other nanoparticle systems. The approach is different from previous nanoparticle patching approaches as iodine is used as a masking agent.

C. Data & methodology: validity of approach, quality of data, quality of presentation

The manuscript is very well written with a concise and systematic presentation of the various steps in the patching process supporting the validity of the approach, showing a deep understanding of the mechanistic details, and an excellent command of characterization and simulation tools. The data is of superb quality and the aesthetic of the figures is very pleasing.

D. Appropriate use of statistics and treatment of uncertainties

The authors use histograms to show patch distributions and %yield to indicate the quality of their patching process. It would be useful if the authors would define what is meant by “achieving the synthesis of more than 20 different patchy NPs in high yield (Fig1, Supplementary Figs. 3-15)” besides “simply” showing TEM images of a few particles per synthesis. Maybe the yield could be expressed in terms of the amount of gold nanoparticle seeds used and patchy particles made.

E. Conclusions: robustness, validity, reliability

The authors present convincing arguments that their patching approach is robust for gold nanoparticles of various geometries and that patches are formed in reliable fashion. It is not apparent how easy this patching process can be translated to other nanoparticle systems as the authors claim in the conclusion “This experiment-theory-simulation approach can immediately be extended to other NP systems,…” Gold is quite special in its chemical properties and binding interactions with other materials, which is likely the reason why it was chosen by the authors in the first place.

F. Suggested improvements: experiments, data for possible revision

A few revisions are recommended as described below, but no additional data or experiments are needed.

G. References: appropriate credit to previous work?

The authors have cited relevant literature and given credit to previous work.

H. Clarity and context: lucidity of abstract/summary, appropriateness of abstract, introduction and conclusions

The authors have provided a well-written manuscript that is densely packed with a large amount of insightful information and delightful results.

Comments to improve the manuscript:

Page 3 Line 102, “With these two important steps demonstrated, we explore…”

Please, define “high yield” in this sentence. Figure 1 and the supporting info only supply images of a few particles that show patch uniformity, but do not provide information about the yield.

Figure 1. It seems that w/o iodine row should come after the w/iodine row to follow the sequence in the manuscript text.

Page 6 Lines 200-201, “We do observe configurations in which 2-NAT is partially…”

Please, explain why it is presumed that the physisorbed 2-NAT will be washed off.

Figure 2e Readability of the $\Delta\mu_{2-NAT}$ vs μ_I plot could be improved. Maybe it would be easier to mark the smaller fields with numbers and then put a list of the numbers and the corresponding three-color bar images in the big white space on the bottom right or maybe connect the color bars with lines to the fields they refer to.

Figure 4 Caption. Please, define Ω also in the caption.

Page 11 lines 349-353, “Usage of T …”

This sentence is not clear and could be improved.

Page 14, Conclusion, lines 433-436

Gold seems to be rather unique in its ability to support reversible binding and not react with surface ligands or adatoms. It would be helpful if the authors could provide at least one other example system where they predict a similar approach could work. If such a system is not easy to identify, maybe the authors could at least provide guidance on what characteristics such a system would have to exhibit.

Extended Data Fig 2.

The variability of the gold nanoparticle shapes in this figure is very surprising and requires at least a comment as to why this data is shown. The synthesis seems to involve nanorods followed by etching processes leading to very uniform gold nanoparticle seeds as shown in the supporting info, but ED Figure 2 shows the often-encountered variability in gold nanoparticle shapes.

General comment:

It is somewhat surprising that the assembly experiments are only shown for the most symmetrically modified patches. Although this may be outside the scope of this already very dense paper, curiosity drives the following questions: Have the authors tried assembly of the non-symmetrically patched nanoparticles? Were difficulties encountered in these systems? Or do the calculations predict interesting structures for asymmetric patch geometries?

We thank all the three reviewers' great comments, which have made our manuscript much stronger. Below we detail our point-to-point response and revisions (text changes quoted and colored **blue**). Revisions include new experiments, analysis, Monte Carlo simulations, references, and discussions as changes in the **main text**; revised **Fig. 2**, revised **Method Section**, revised **Supplementary Note 6**, revised **Supplementary Figs. 4, 13, 15, 16, 30, 31, and 59**, revised **Supplementary Tables 2, 6, and 7**; new **Supplementary Notes 2.8 and 2.9**, new **Supplementary Figs. 17, 18, 37, and 60**, new **Supplementary Table 1**.

Responses to Reviewer 1: Pages R2 to R9

Responses to Reviewer 2: Pages R10 to R19

Responses to Reviewer 3: Pages R20 to R28

Responses to Reviewer 1

Overall Comment: “*This manuscript presents a significant advancement in the bottom-up fabrication of precisely patterned nanoparticles (NPs) at nanoscale. The authors report a robust method for producing uniform and diverse gold patchy nanoparticles using atomic-scale “stenciling” with halide ions, specifically iodide, followed by ligand-mediated polymer grafting. The “stenciling” process is finely tunable by modulating the structure of iodide submonolayers through simple variation of iodide ion concentration. Transmission electron microscopy (TEM) combined with density functional theory (DFT) calculations indicates that patch formation arises from the selective adsorption of iodide ions onto (111) facets of gold nanoparticles at low iodide concentrations, leaving the unmasked regions available for polymer grafting. This facet-selective masking was convincingly demonstrated across a variety of gold nanoparticle morphologies, including octahedra, cuboctahedra, rhombic dodecahedra, cubes, and bipyramids. Furthermore, these well-defined patchy nanoparticles can self-assemble into superlattices, highlighting their potential as versatile building blocks in nanostructured materials. Overall, this is an elegant and practical approach for creating uniform patchy nanoparticles with promising applications in targeted delivery, catalysis, self-assembly, and nanomaterials engineering. I recommend publication of this manuscript following minor revision.*”

Reply: We greatly appreciate the positive and constructive comments from the reviewer, which has made our manuscript stronger! As detailed below, we have addressed all the reviewer’s suggestions with new experiments on successful extension of the stenciling method to nanoparticles (NPs) of other composition and more detailed controls of the masking processes, as well as new discussions.

Comment 1: “*The manuscript describes the successful synthesis of many distinct types of patchy nanoparticles and demonstrates that iodide ions preferentially adsorb on the (111) facets of gold nanoparticles under low-concentration conditions. While those results are absolutely fascinating, will the decrease of 2-NAT give similar outcomes? A minor thing: the authors show patch formation on gold nanoprisms at their tips—does this suggest broader applicability to other anisotropic shapes such as nanorods?*”

Reply: We thank the reviewer for the great comments! We have performed the following new experiments.

First, without iodides added, we decreased [2-NAT], using gold octahedra as an example. As shown below in the new **Supplementary Fig. 4c,d**, we observed a transition from fully coated NPs to almost random, sparsely coated NPs as [2-NAT] decreases. At low [2-NAT], the polymers can no longer fully coat the NP surface due to insufficient 2-NAT coverage, inducing sparse patches. Meanwhile, the 2-NAT coverage also loses its facet selectiveness due to the absence of iodide masking. As a result, small patches randomly appear both on the {111} face and {100} tip facets of octahedra NPs, sometimes with multiple patches even on one face. There is visibly large polydispersity in patch shape, pattern, and sizes. This control experiment confirms again the critical role of the iodide masking in generating predictive and precise patch patterns.

Second, about anisotropic NPs other than regular polyhedral shapes, both the elongated bipyramid NPs in our original manuscript (e.g., **Figure 1c, column 2; Figure 3f**) and our new experiments on gold nanorods (**Figure R1** below) show that selective stenciling still works. Specifically, for the bipyramid NPs, because their faces are composed of {111} facets, all faces are masked by iodide, resulting in tip and equator patches (**Figure 1c, column 2**). In our new experiments on gold nanorods, [Redacted]

[Redacted] they also show highly monodisperse side facet patches. These rods were synthesized following the method presented in *Nano Lett.* 13, 765 (2013), with a size of ~90 nm in length and ~38 nm in width. The rods of such size tend to have alternating {110} and {100} facets on the side, which agrees with our observation of alternating side facet patches because {110} can be more easily masked than {100}. Meanwhile, rods of different sizes synthesized from different recipes can exhibit very rich surface facets, leading to a huge parameter space of patch design with the stenciling strategy. We thus see patching of gold rods belongs to a different paper of its own.

[Redacted]

In the revised manuscript, we have made the following changes to address the comments.

1. We have added new supplementary figures of new experiments to show that (i) without iodide added, decreasing [2-NAT] leads to discrete polymer domains but without precision of the patching facets or patch uniformity (new **Supplementary Fig. 4c,d**), and (ii) with iodide, decreasing [2-NAT] would cause shrinking of patches (new **Supplementary Fig. 13e**). The details of new experiments have been added in **Supplementary Tables 2 and 7**.

Supplementary Fig. 4c,d. 2-NAT concentration effect on patch patterns of gold octahedra. (c) Representative TEM images of the control experiment on polymer-coated octahedra synthesized using varying [2-NAT] without iodide masking. (d) Low-magnification TEM images of the polymer-coated octahedra. In c, polymer patches are false-colored in cyan. For detailed synthesis conditions, see **Supplementary Note 2.1** and **Supplementary Table 2**. Scale bars: (c) 20 nm and (d) 100 nm.

Supplementary Fig. 13e. Synthesis of patchy large rhombic dodecahedra. (e) Representative SEM images of patchy large rhombic dodecahedra synthesized using varying [2-NAT] at $[I^-]$ fixed of 8.33 μM (inset: corresponding schematic of patchy NPs). As [2-NAT] increases, the twelve blue face patches expand their size, merge together, and web-like patch pattern is formed. For synthesis conditions, see **Supplementary Note 2.6** and **Supplementary Table 7**. Scale bars: 50 nm.

Supplementary Table 2. Reaction conditions for synthesizing patchy octahedra with various patch patterns.

Masking step*	Polymer grafting step**			Figure index
	Volume of DMF (μL)	Concentration of 2-NAT solution (mg/mL)	Volume of 2-NAT solution (μL)	
Final iodide concentration for NP incubation (μM)				
0	815	0.002	5	Supplementary Fig. 4c,d
0	815	0.02	5	Supplementary Fig. 4c,d
0	815	0.2	5	Supplementary Fig. 4c,d
0	800	0.2	20	Supplementary Fig. 4c,d

* During the masking step, the volume of Stock solution I (0.5 OD at λ_{max}) is fixed as 6.9 mL.

** Note that during the polymer grafting step, fixed volumes of 80 μL of PS-*b*-PAA solution (8 mg/mL in DMF) and 200 μL of Stock Solution II (5.0 OD at λ_{max} , 0.07 mM CTAB) are added to the reactor. The polymer grafting reaction is done at 110 °C.

Supplementary Table 7. Reaction conditions for synthesizing patchy large rhombic dodecahedra with various patch patterns.

Masking step*	Polymer grafting step**			Figure index
	Volume of DMF (μL)	Concentration of 2-NAT solution (mg/mL)	Volume of 2-NAT solution (μL)	
Final iodide concentration for NP incubation (μM)				
8.26	720	0.02	100	Supplementary Fig. 13c

* During the masking step, the volume of Stock solution I (0.5 OD at λ_{\max}) is fixed as 7.2 mL.

** Note that during the polymer grafting step, fixed volumes of 80 μL of PS-*b*-PAA solution (8 mg/mL in DMF) and 200 μL of Stock Solution II (5.0 OD at λ_{\max} , 0.007 mM CTAB) are added to the reactor. The polymer grafting reaction is done at 110 °C.

2. We added discussions in the main text,

“When $[\text{I}^-]$ is fixed, $[\text{2-NAT}]$ variations change the patch size (Supplementary Fig. 4a,b). In our control experiment without iodide masks, decreasing $[\text{2-NAT}]$ causes insufficient 2-NAT coverage and incomplete polymer coating domains on NP surfaces (Supplementary Fig. 4c,d). These polymer domains do not exhibit facet selectivity and are polydisperse in their size, shape, and pattern, showing the importance of iodide masks in the precise control of patch patterns.”

“At relatively low $[\text{I}^-]/[\text{2-NAT}]$, only the $\{111\}$ surfaces are masked, leading to a complex hybrid patch pattern: extended patches on the $\{110\}$ faces merging into web-like patches with tuneable widths, together with small patches, each of a diameter of 9.0 ± 1.0 nm, precisely located on the six $\{100\}$ vertices (Fig. 3q, Supplementary Fig. 28, Supplementary Fig. 13a,b,e, Supplementary Table 8). At this condition, the eight $\{111\}$ vertices remain polymer-free. At high $[\text{I}^-]/[\text{2-NAT}]$, the six $\{100\}$ vertices are next to be masked, in agreement with the DFT calculations, leading to only face patches as all vertices are now masked (Fig. 3r, Supplementary Fig. 13c–e).”

“For example, as demonstrated with bipyramid NPs, the stenciling strategy can be applied to highly anisotropic, elongated NP shapes. Gold nanorods are thus promising candidates worth further study due to their rich faceting behaviours determined by their size and synthesis conditions⁴⁶.”

Comment 2: “Will the selective adsorption of iodide ions be unique to gold? Could this “stenciling” strategy be extended to other noble or coinage metals (e.g., silver or copper) with comparable surface chemistries? If so, do iodide ions also preferentially mask the $\langle 111 \rangle$ facets of these metals, or is the observed selectivity specific to gold-iodide coordination chemistry?”

Reply: Thank you for the great comment!

In the revised manuscript, we have added new experiments to show the success of applying the stenciling strategy to palladium (Pd) nanocubes using iodides as masks, producing highly uniform face patches (Supplementary Fig. 60). The foundational principle of stenciling is to use facet-selective adsorption of species (e.g., atoms, ions, ligands) to mask surface sites to allow only the unmasked sites to be further coated. As long as this principle can work, stenciling can work for other compositions of metal NPs, though not necessarily always masking $\{111\}$ or using iodides. In the case of Pd, first-principles studies using density functional theory (DFT) indicate that iodide binds more strongly to Pd(111) than to Au(111) (*Phys. Chem. Chem. Phys.* 16, 13630 (2014); *J. Electrochem. Soc.* 163, H796 (2016)) because iodide is more polarized on Pd. Moreover, Pd has a smaller lattice constant than Au, which would enhance its masking effect by steric repulsion. Relatedly, iodide can also form into dense layers on Pd(110) as discussed in Ref. (*J. Phys. Chem. C* 118, 29919 (2014)), which could preclude 2-NAT adsorption onto the Pd(110) surface.

Consistent with the above literature, both the edge Pd(110) and vertex Pd(111) facets in Pd nanocubes show no coating of polymers in our experiment (Supplementary Fig. 60). Iodide adsorption on Pd is generally less studied than that on Au and previous literature does not consider co-adsorption of iodide and 2-NAT as in our work. More detailed DFT analysis is needed to fully understand the mechanism of stenciling for other metal NPs or stencils, which goes beyond the scope of this first study.

In the revised manuscript, we have made the following revisions to address this comment.

1. We added a new **Supplementary Fig. 60** summarizing our experiment of making Pd patchy nanocubes.

Supplementary Fig. 60. Synthesis of patchy palladium nanocubes. (a) Schematic illustrating the synthesis of palladium cubes. Iodides are introduced during the palladium nanocube synthesis, intrinsically enabling selective facet masking for polymer grafting. (b) EDX mapping of the palladium cubes showing composition uniformity. (c–l) High-angle annular dark field (HAADF)-STEM images of as-synthesized palladium cubes and corresponding 3D schematic. (d,i) show the viewing direction of the STEM images. (e–g) and (j–l) are zoomed-in views of the NP regions boxed in (c) and (h), showing that the vertices, edges, and faces exhibit $\{111\}$, $\{110\}$, and $\{100\}$ facets, respectively, as noted by the yellow lines. (m,n) Representative SEM images of patchy palladium cubes. Inset: TEM image (m) and corresponding 3D schematic (n) of a patchy cube with cyan colored region as polymer patches. For these samples, iodides exist as an additive during the palladium cube synthesis process as shown in a. The as-synthesized NPs are used directly for polymer grafting without additional iodide masking step. (o,p) Representative SEM image of patchy palladium cubes synthesized with the additional iodide masking step performed on the as-synthesized palladium cubes (o) and SEM image with selected particles false-colored to highlight the patches (cyan). As more iodide is added, the patch sizes decrease, consistent with the general trend of increased masking at higher $[I^-]$ during stenciling (as seen from m,n to o,p). See **Supplementary Note 2.8** for synthesis details. Scale bars: (b) 50 nm, (c,h) 20 nm, (e–g, j–l) 2 nm, and (m–p) 100 nm (inset: 20 nm).

- We added a new **Supplementary Note 2.8** on the experimental details of patchy palladium nanocube synthesis.

“2.8. Synthesis of patchy palladium nanocubes

Palladium nanocubes are synthesized following a previously reported method with slight modifications⁷. 20 mM palladium(II) chloride (H_2PdCl_4) is first prepared by dissolving 11.78 mg of

sodium tetrachloropalladate(II) (Na_2PdCl_4) in 2 mL of 40 mM HCl. The solution is tightly capped and kept undisturbed in a water bath at 30°C for 1 h. Then, 625 μL of the prepared 20 mM H_2PdCl_4 is added to 6.25 mL of 100 mM CTAB in a 20 mL vial, followed by sequential addition of 1.25 mL of 40 mM potassium iodide (KI) 0.5 mL of 100 mM AA, and 0.75 mL of DI water under shaking at 400 rpm. The vial is capped and quickly transferred to an oil bath, followed by heating at 90°C for 1 h with stirring at 400 rpm. Afterward, the reaction solution is cooled by immersion in a water bath. To remove unreacted reactants, the as-synthesized palladium cube solution is transferred into a 15 mL centrifuge tube and centrifuged twice at 8,500 rpm for 10 min each. After the first centrifugation, the supernatant is removed, and the pellet is redispersed in 5 mL of DI water. Following the second round, the supernatant is removed again, and the pellet is dispersed in 2.5 mL of 20 mM CTAB.

Patchy palladium cubes shown in **Supplementary Fig. 60m,n** are synthesized following the same general procedure as described in Supplementary Note 2.1, but with adjusted [2-NAT] and without the iodide masking step. Specifically, 213 μL of the palladium cube solution from above is transferred into a 1.5 mL microcentrifuge tube and centrifuged at 5,500 rpm for 15 min. The 10 μL pellet is redispersed in a 15 mL centrifuge tube using 20 mM CTAC to reach a total volume of 7.2 mL. The solution is then centrifuged twice. After the first centrifugation at 7,600 rpm for 15 min, the pellet is redispersed in 10 mL of 20 mM CTAC and centrifuged again at 7,600 rpm for 15 min. After the second centrifugation, the pellet is redispersed in 1 mL of DI water and transferred into a 1.5 mL microcentrifuge tube, followed by a final centrifugation at 3,250 rpm for 15 min. After removing the supernatant, the final CTAB concentration and volume are adjusted to 0.07 mM and 610 μL , respectively, using additional DI water and 0.16 mM CTAB solution. For polymer grafting, 815 μL of DMF is first added to an 8 mL vial. Then 5 μL of 2-NAT solution (0.02 mg/mL in DMF), 200 μL of the adjusted palladium cube solution, and 80 μL of PS-*b*-PAA solution (8 mg/mL in DMF) are sequentially mixed by dropwise addition under mild vortex. The vial is tightly capped with a Teflon-lined cap, sonicated for 5 s, sealed with parafilm, heated at 110 °C in an oil bath, and left undisturbed for 2 h. The reaction mixture is then cooled down to RT in the oil bath, which typically takes 90 min. The solution is transferred to 1.5 mL centrifuge tubes and centrifuged three times at 4,500 rpm, 2,250 rpm, and 1,750 rpm for 15 min each, to separate the residual 2-NAT and PS-*b*-PAA from the patchy Pd NPs. After the first and second centrifugations, 1.49 mL of the supernatant is removed, and the 10 μL pellet is redispersed with 1.49 mL of water. Following the third round, the 10 μL pellet is diluted with 490 μL of water for long-term storage for TEM observations. Patchy palladium cubes shown in **Supplementary 60o,p** are synthesized the same as above, except that the as-synthesized palladium nanocubes undergo an additional iodide masking step with a final [NaI] concentration of 6.62 μM .”

3. We added additional discussions on the new experiments and more literature in the main text.

“It can also be applied to other metal NPs such as palladium nanocubes (**Supplementary Fig. 60**), where the presence of iodide leads to the formation of face-patched palladium nanocubes, similar to the face-patched gold nanocubes. Iodide adsorption on Pd(111) and Pd(110) has been discussed in previous literature^{47,48}, though the specific 2-NAT co-adsorption mechanism needs further study. The general principle of facet-selective adsorption and masking is likely to be generalizable to other systems. For example, DFT calculations and experiments show a potential masking effect of iodide or chloride for hexadecylamine (HDA) adsorption on Cu surfaces, in that the {100} facets are covered with halide and HDA, while the {111} facets contain only halide in a select range of solution-phase chemical potentials^{43,49,50}. Similarly, iodides can also bind strongly to Ag(111) and Ag(110), suggesting potentially a similar masking effect to that on gold⁵¹.”

Comment 3: “During nanoparticle synthesis, halide ions such as Cl^- and Br^- are commonly introduced. What is their fate post-synthesis, and do they compete with or influence the adsorption behavior of iodide ions? What role do other surface capping ligands play in the adsorption behavior? For instance, where would cetyltrimethylammonium bromide be after the formation of polymer domains.”

Reply: Thank you for this great point! Indeed, halide ions such as Cl^- and Br^- can be carried over from the gold NP synthesis and/or introduced as the surfactants to disperse NPs in the iodide masking and polymer grafting steps. Yet they exhibit negligible impact on the iodide masking. Among different halide ions (e.g., chloride, bromide, iodide), iodide exhibits the strongest binding affinity to gold. This has been proved in our control experiments as Cl^- and Br^- alone cannot achieve facet masking (**Supplementary Fig. 16c–e**). In our regular iodide masking step, though we disperse pristine NPs in cetyltrimethylammonium bromide (CTAB) or cetyltrimethylammonium chloride (CTAC) to prevent NP aggregation, iodides can still bind to the gold surface for polymer patching. We also show in our new control experiments (new **Supplementary Fig. 16a,b**) that over a broad range of [CTAB] and [CTAC] used to disperse NPs, patchy NP synthesis has been robustly achieved. In the polymer grafting step, the final [CTAB] is kept low, below 0.1 mM. At this concentration, CTAB has negligible influence on polymer grafting but stabilizes, from our empirical observation, the gold NPs during the initial grafting process. In contrast, if the final [CTAB] is high during polymer grafting, we anticipate large aggregations of patchy NPs resulting from surfactant–polyelectrolyte (PS-*b*-PAA) complexation, as inferred from our previous work (*ACS Nano* 18, 939 (2024)) and also comproportionation reaction-induced etching of core NPs (*J. Am. Chem. Soc.* 136, 7603 (2014)). In our experiment, we do not observe NP aggregation or etching of core NPs. Given that the final [CTAB] is well below the critical micelle concentration of CTAB, we expect the CTAB to either freely disperse in solution after the formation of polymer domains or mildly adsorb onto the negatively charged PAA blocks of the patches due to electrostatic attraction (*ACS Nano* 18, 939 (2024)).

In the revised manuscript, we have made the following revisions to address the comment.

1. We included an updated **Supplementary Fig. 16** to show that CTAB and CTAC solutions of varied concentrations used for dispersing NPs in masking step do not impact the polymer patching behaviors.

Supplementary Fig. 16. Control studies on surfactant conditions and halide ion specificity in masking. (a,b) Representative TEM images of patchy octahedra with iodide masking performed in different surfactant environments: CTAC (a) and CTAB (b) at varying concentrations. The same vertex-patched octahedra are synthesized regardless of the surfactant type or concentration during iodide masking. $[\text{I}^-]$ and $[\text{2-NAT}]$ are fixed at $6.62 \mu\text{M}$ and $17.0 \mu\text{M}$, respectively. (c–e) Representative TEM and SEM images of polymer-coated octahedra synthesized after incubation with different halide ions: chloride (c), bromide (d), and iodide (e). Vertex-patched octahedra are synthesized only when incubated with iodide. The concentrations of the halide ions and 2-NAT are fixed at $0.25 \mu\text{M}$ and $17.0 \mu\text{M}$, respectively. Scale bars: 100 nm (inset: 20 nm).

- We added additional discussions in the main text:

“Although other halide ions such as bromide and chloride are present during iodide masking, they show negligible effects on iodide adsorption and polymer patching (Supplementary Fig. 16a,b). As expected, control experiments using bromide and chloride as masking agents generate non-patchy NPs (Methods, Supplementary Fig. 16c–e).”

Responses to Reviewer 2

Overall Comments: *“The authors present a novel bottom-up approach to nanoparticle (NP) surface patterning, termed “atomic stencilling”, which uses iodide adsorption as a molecular mask and ligand-mediated polymer grafting as a “paint” to create patchy nanoparticles. This method enables the synthesis of over 20 distinct patchy NP morphologies with high yield and precision. Authors use a combination of density functional theory (DFT), molecular dynamics (MD), and Monte Carlo (MC) simulations to explain and predict patch formation and NP self-assembly into non-close-packed superlattices. The co-adsorption of iodide and 2-naphthalene thiol on gold depends on the type of gold facet, which they explain using DFT, leading to selective surface masking that allows us to achieve various patch patterns. Then, they “paint” the iodide-masked NPs by ligand-mediated polymer grafting, in which polymers selectively adhere only to the unmasked gold surfaces, rendering patches.”*

“This work is original and a novel contribution to the field of nanoparticle surface engineering and self-assembly. The atomic stencilling approach is both conceptually elegant and experimentally robust, offering a new paradigm for creating anisotropic, functional nanomaterials. This work has major implications across nanotechnology, materials science, and applied physics.”

“Specific parts that I consider original (and cool!): The idea of using atomic-scale masking via iodide adsorption is highly original and bridges a gap between top-down lithography and bottom-up self-assembly. The ability to control NP surface chemistry at the nanometer scale opens new avenues in colloidal crystal engineering, metamaterials, drug delivery, etc. Demonstration of millimeter-scale superlattices with BCC and BCT symmetry is a significant result!”

Reply: We greatly appreciate the greatly supportive comments and the great suggestions from the reviewer, all of we have addressed by adding new experiments, new Monte Carlo (MC) simulations, new discussions, and new references. We thank the reviewer for helping us make our work more thorough and stronger.

Comment 1: *“Their approach is a synergistic integration of synthesis, advanced characterization (e.g., STEM-EDX, XPS, Raman, tomography), and multiscale modeling. I don't know how much “polymer theory” helped here, so I would suggest they remove that. But the use of DFT to rationalize facet-selective iodide adsorption and its impact on ligand binding is rigorous and well-supported. Computational prediction of patch morphology and self-assembly behavior also a strength. Phase diagrams and simulations provide a powerful tool for designing future patchy NPs. Conclusions seem robust and justified.”*

Reply: We thank the reviewer’s appreciation of the importance of our density functional theory (DFT), MC simulation of self-assembly, and phase diagram of patching behaviors in understanding and predicting the formation of patchy nanoparticles (NPs). We hope to note that the polymer scaling theory presented in this work provides an indispensable bridge connecting the DFT (atomistic) calculations of iodide adsorption to the mesoscale particle features such as patch shape, pattern, and size presented in the phase diagram (**Figure 4i**). These particle features serve as the foundation for the self-assembly simulation (**Figure 5**). Specifically, DFT alone cannot yield information on the patches because the patches are not short alkanes but block copolymer chains. The chains experience microphase separation intrinsic to block copolymers and possess sizes similar in magnitude to the edge lengths of the core NPs. As such, polymeric interactions contribute a significant amount of additional entropic and enthalpic effects that fine-tune the patching behaviors as summarized in **Figure 4i**.

In the revised manuscript, we have added discussions in the main text to further clarify the role of polymer scaling theory,

*“While iodide masking dictates the gold NP facets available for polymer grafting and thus patch patterning, our patches are PS-*b*-PAA block copolymers, which experience intrinsic microphase separation. Furthermore, these polymers possess sizes similar in magnitude to the size of the core NPs. As such, we expect that enthalpic and entropic effects associated with the polymers further fine-tune, within the set of available grafting sites, the patch patterns on NPs and the size of patches^{29,38}. To*

elucidate these effects, we develop a multiscale modelling workflow to bridge the atomistic-level, electronic DFT model with mesoscale polymer behaviour. We first construct a statistical mechanical adsorption model⁴⁵ for iodide masking that uses the binding energies computed by DFT, which determines the stoichiometric ratios of surface binding sites available for polymer grafting on each facet type of the NP (**Methods, Supplementary Fig. 30, Supplementary Note 5**). In this way, we directly map DFT calculations of individual facets to the 3D NP shapes, thereby quantifying how surface masking affects the thermodynamics that govern chain conformations (**Fig. 4a**). Microscopically, neighbouring grafted polymers induce crowding, causing chain extension and a large entropic penalty²⁹. At the same time, chain extension also facilitates PS–PS contacts, which is enthalpically favourable (**Supplementary Note 6**). This competition between conformational entropy and inter-chain attraction E_{chain} determines the free energy of a grafted polymer chain at the unmasked NP surface locations.”

Comment 2: “While the method is elegant, the scalability of the iodide masking and polymer grafting process is not addressed. Maybe I missed this but what about the sensitivity to reaction conditions (e.g., iodide concentration, temperature, CTAB levels); could that limit reproducibility outside controlled lab settings.”

Reply: We thank the reviewer for the great comment. In the revised manuscript, we added new experiment results (new **Supplementary Fig. 17**) to show the successful synthesis of patchy NPs with greatly increased reaction volumes (as high as 20×), characterized with transmission electron microscopy (TEM) and scanning electron microscopy (SEM) imaging. This much volume increase in fact requires a completely different reaction setup, from our small vials to a more standardized 50 mL round-bottom flask. Because the iodide masking is a quite robust step, we anticipate that the patchy NP synthesis will be reproducibly scalable as long as (1) the reaction parameters (e.g., [I⁻], [2-NAT], [CTAB]) follow the methods described in our manuscript, and (2) the reaction environment (e.g., temperature) is well controlled to ensure homogeneous reaction.

In the revised manuscript, we have added the new results and experimental details to address the comments as below.

1. We added a new **Supplementary Fig. 17** to show our scale-up synthesis using the patchy octahedra as an example. All the details of the new experiments are added in **Supplementary Table 2**.

Supplementary Fig. 17. Scale-up synthesis of patchy gold octahedra. (a) TEM images of patchy octahedra synthesized at different reaction volumes scaled up from the standard procedure described in

Supplementary Note 2.1. (b) Photograph of the reaction setup for a 20-fold scale-up reaction of patchy octahedra synthesis. (c) Representative TEM images of patchy octahedra synthesized at the 20-fold scale (inset: high-magnification TEM image). Vertex-patched octahedra are obtained consistently, showing that our atomic stenciling method is scalable. Scale bars: (a) 100 nm, (b) 1 cm, and (c) 200 nm (inset in c: 20 nm).

Supplementary Table 2. Reaction conditions for synthesizing patchy octahedra with various patch patterns.

Masking step*	Polymer grafting step**			Figure index
Final iodide concentration for NP incubation (μM)	Volume of DMF (μL)	Concentration of 2-NAT solution (mg/mL)	Volume of 2-NAT solution (μL)	
6.62****	1630	2	10	Supplementary Fig. 17a 2 \times scale
6.62****	3260	2	20	Supplementary Fig. 17a 4 \times scale
6.62****	8150	2	50	Supplementary Fig. 17a 10 \times scale
6.62****	16350	2	50	Supplementary Fig. 17b,c

**** For scaled-up synthesis, the synthesis protocol is slightly modified. For detailed synthesis conditions, see **Supplementary Note 2.9**.

2. We added a new **Supplementary Note 2.9** on the details of the scale-up reaction.

“2.9. Large-scale synthesis of patchy gold octahedra

The large-scale synthesis of patchy gold octahedra is performed following the recipe described in **Supplementary Note 2.1** with the total reaction volumes increased. The experimental results and the synthesis conditions are summarized in **Supplementary Fig. 17** and **Supplementary Table 2**. Take the 20-fold scaled-up synthesis of patchy octahedra as an example, seven duplicates of iodide masked gold octahedra solutions as described in **Supplementary Note 2.1.1** are prepared. After the last step of centrifugation and removal of supernatant, the pellets are combined into one 15 mL centrifuge tube, with the final CTAB concentration and NP concentration adjusted to 0.007 mM and 5 OD at λ_{max} , respectively, using additional DI water and 0.16 mM CTAB. For this step, we use the duplicates of small-volume reactions to ensure a good control of the degassing and mixing process. This step can be scaled up potentially as long as good degassing and mixing can be achieved in a big flask.

For the polymer grafting step, direct scaling up by 20 times in a 50 mL round-bottom flask is achieved. Specifically, 16.35 mL of DMF is first added to a 50 mL round-bottom flask. Then, 50 μL of 2-NAT solution (2 mg/mL in DMF), 4 mL of the gold octahedron solution prepared after iodide masking, and 1.6 mL of PS-*b*-PAA solution (8 mg/mL in DMF) are sequentially added into the flask dropwise under sonication for about 20 s to ensure good mixing. After sealing the flask with a glass stopper, wrapping the flask neck with parafilm, and securing it with a flask clip, the flask is heated at 110 $^{\circ}\text{C}$ in an oil bath, and left undisturbed for 2 h. The reaction mixture is then cooled down to RT in the oil bath, which typically takes 90 min. For TEM and SEM characterizations, 1 mL of the reaction mixture was washed by three rounds of centrifugation at 4,500 rpm for 15 min, 2,500 rpm for 15 min, and 1,750 rpm for 15 min each, to separate the residual 2-NAT and PS-*b*-PAA from the product patchy octahedra. After the first and second centrifugations, 1.49 mL of the supernatant is removed, and the

10 μL pellet is redispersed with 1.49 mL of water. Following the third round, the 10 μL pellet is diluted with 490 μL of water for long-term storage.”

3. We added a discussion in the main text about the scalability of the stenciling method,

“The stencilling method is scalable (**Supplementary Fig. 17, Supplementary Note 2.9**) and effective for other thiols and block copolymers (**Supplementary Fig. 18**).”

Comment 3: “*The study focuses exclusively on gold nanoparticles. Can the authors comment on how extendable this idea of atomic stencilling is to other materials (e.g., silver nanoparticles)?*”

Reply: Thank you for the great comment! As the reviewer commented, it will be a great demonstration of the potency of our stenciling method if we can extend it to other compositions of metal NPs. In the revised manuscript, we added new experimental results to show the success of applying the stenciling strategy to palladium nanocubes using iodides, producing highly uniform face patches (**Supplementary Fig. 60**). The foundational principle of stenciling is to use facet-selective adsorption of species (e.g., atoms, ions, ligands) to mask surface sites to allow only the unmasked sites to be further coated. As long as this principle works, stenciling can work for other compositions of metal NPs, though not necessarily always masking $\{111\}$ or using iodides.

Considering extension to silver NPs, we have done additional literature search and found that the adsorption of iodide has been studied experimentally and with DFT on Ag(111) (*J. Phys. Chem.* 156, 164702 (2022)), Ag(110) (*Surf. Sci.* 760, 122769 (2025)), and Ag(100) (*Phys. Rev. B* 80, 125409 (2009)). There was also an experimental study comparing iodide adsorption on all the three Ag surfaces (*Surf. Sci.* 128, 145 (1983)). Iodide forms a relatively open $c(2\times 2)$ pattern with a coverage of 1/2 on Ag(100), while it can form compressed structures, with a saturation coverage—the highest iodide coverage below intermixing—of 0.69 on Ag(110) and 0.38 on Ag(111). The relatively high saturation coverages on Ag(110) and Ag(111), combined with the relatively strong binding of iodide to Ag(111) compared to Au(111) (*J. Electrochem. Soc.* 163, H796, 2016) seem to point to a similar masking effect to that seen on gold.

We have also done literature search to rationalize our observation of patchy palladium nanocubes with face patches. First-principles studies using DFT indicate that iodide binds more strongly to Pd(111) than to Au(111) (*Phys. Chem. Chem. Phys.* 16, 13630 (2014); *J. Electrochem. Soc.* 163, H796 (2016)) because iodide is more polarized on palladium. Moreover, palladium has a smaller lattice constant than gold, which would enhance its masking effect by steric repulsion. Relatedly, iodide can also form into dense layers on Pd(110) as discussed in Reference of *J. Phys. Chem. C* 118, 29919 (2014), which could preclude 2-NAT adsorption onto the Pd(110) surface. Consistent with the literature, both the edge Pd(110) and vertex Pd(111) facets in palladium cubes show no coating of polymers in our experiment (**Supplementary Fig. 60**). Iodide adsorption on palladium is generally less studied than that on gold. Previous literature also does not consider co-adsorption of iodide and 2-NAT, like what we did in our work. More detailed DFT calculation and analysis are needed to fully understand the mechanism of stenciling for other metal NPs or stencils.

In the revised manuscript, we have made the following revisions to address this comment.

1. We added a new **Supplementary Fig. 60** summarizing our experiment of making Pd patchy nanocubes.

Supplementary Fig. 60. Synthesis of patchy palladium nanocubes. (a) Schematic illustrating the synthesis of palladium cubes. Iodides are introduced during the palladium nanocube synthesis, intrinsically enabling selective facet masking for polymer grafting. (b) EDX mapping of the palladium cubes showing composition uniformity. (c–l) High-angle annular dark field (HAADF)-STEM images of as-synthesized palladium cubes and corresponding 3D schematic. (d,i) show the viewing direction of the STEM images. (e–g) and (j–l) are zoomed-in views of the NP regions boxed in (c) and (h), showing that the vertices, edges, and faces exhibit $\{111\}$, $\{110\}$, and $\{100\}$ facets, respectively, as noted by the yellow lines. (m,n) Representative SEM images of patchy palladium cubes. Inset: TEM image (m) and corresponding 3D schematic (n) of a patchy cube with cyan colored region as polymer patches. For these samples, iodides exist as an additive during the palladium cube synthesis process as shown in a. The as-synthesized NPs are used directly for polymer grafting without additional iodide masking step. (o,p) Representative SEM image of patchy palladium cubes synthesized with the additional iodide masking step performed on the as-synthesized palladium cubes (o) and SEM image with selected particles false-colored to highlight the patches (cyan). As more iodide is added, the patch sizes decrease, consistent with the general trend of increased masking at higher $[I^-]$ during stenciling (as seen from m,n to o,p). See **Supplementary Note 2.8** for synthesis details. Scale bars: (b) 50 nm, (c,h) 20 nm, (e–g, j–l) 2 nm, and (m–p) 100 nm (inset: 20 nm).

- We added a new **Supplementary Note 2.8** on the experimental details of patchy palladium nanocube synthesis.

“2.8. Synthesis of patchy palladium nanocubes

Palladium nanocubes are synthesized following a previously reported method with slight modifications⁷. 20 mM palladium(II) chloride (H_2PdCl_4) is first prepared by dissolving 11.78 mg of

sodium tetrachloropalladate(II) (Na_2PdCl_4) in 2 mL of 40 mM HCl. The solution is tightly capped and kept undisturbed in a water bath at 30°C for 1 h. Then, 625 μL of the prepared 20 mM H_2PdCl_4 is added to 6.25 mL of 100 mM CTAB in a 20 mL vial, followed by sequential addition of 1.25 mL of 40 mM potassium iodide (KI) 0.5 mL of 100 mM AA, and 0.75 mL of DI water under shaking at 400 rpm. The vial is capped and quickly transferred to an oil bath, followed by heating at 90°C for 1 h with stirring at 400 rpm. Afterward, the reaction solution is cooled by immersion in a water bath. To remove unreacted reactants, the as-synthesized palladium cube solution is transferred into a 15 mL centrifuge tube and centrifuged twice at 8,500 rpm for 10 min each. After the first centrifugation, the supernatant is removed, and the pellet is redispersed in 5 mL of DI water. Following the second round, the supernatant is removed again, and the pellet is dispersed in 2.5 mL of 20 mM CTAB.

Patchy palladium cubes shown in **Supplementary Fig. 60m,n** are synthesized following the same general procedure as described in Supplementary Note 2.1, but with adjusted [2-NAT] and without the iodide masking step. Specifically, 213 μL of the palladium cube solution from above is transferred into a 1.5 mL microcentrifuge tube and centrifuged at 5,500 rpm for 15 min. The 10 μL pellet is redispersed in a 15 mL centrifuge tube using 20 mM CTAC to reach a total volume of 7.2 mL. The solution is then centrifuged twice. After the first centrifugation at 7,600 rpm for 15 min, the pellet is redispersed in 10 mL of 20 mM CTAC and centrifuged again at 7,600 rpm for 15 min. After the second centrifugation, the pellet is redispersed in 1 mL of DI water and transferred into a 1.5 mL microcentrifuge tube, followed by a final centrifugation at 3,250 rpm for 15 min. After removing the supernatant, the final CTAB concentration and volume are adjusted to 0.07 mM and 610 μL , respectively, using additional DI water and 0.16 mM CTAB solution. For polymer grafting, 815 μL of DMF is first added to an 8 mL vial. Then 5 μL of 2-NAT solution (0.02 mg/mL in DMF), 200 μL of the adjusted palladium cube solution, and 80 μL of PS-*b*-PAA solution (8 mg/mL in DMF) are sequentially mixed by dropwise addition under mild vortex. The vial is tightly capped with a Teflon-lined cap, sonicated for 5 s, sealed with parafilm, heated at 110 °C in an oil bath, and left undisturbed for 2 h. The reaction mixture is then cooled down to RT in the oil bath, which typically takes 90 min. The solution is transferred to 1.5 mL centrifuge tubes and centrifuged three times at 4,500 rpm, 2,250 rpm, and 1,750 rpm for 15 min each, to separate the residual 2-NAT and PS-*b*-PAA from the patchy Pd NPs. After the first and second centrifugations, 1.49 mL of the supernatant is removed, and the 10 μL pellet is redispersed with 1.49 mL of water. Following the third round, the 10 μL pellet is diluted with 490 μL of water for long-term storage for TEM observations. Patchy palladium cubes shown in **Supplementary 60o,p** are synthesized the same as above, except that the as-synthesized palladium nanocubes undergo an additional iodide masking step with a final [NaI] concentration of 6.62 μM .”

3. We added additional discussions on the new experiments and more literature in the main text.

“It can also be applied to other metal NPs such as palladium nanocubes (**Supplementary Fig. 60**), where the presence of iodide leads to the formation of face-patched palladium nanocubes, similar to the face-patched gold nanocubes. Iodide adsorption on Pd(111) and Pd(110) has been discussed in previous literature^{47,48}, though the specific 2-NAT co-adsorption mechanism needs further study. The general principle of facet-selective adsorption and masking is likely to be generalizable to other systems. For example, DFT calculations and experiments show a potential masking effect of iodide or chloride for hexadecylamine (HDA) adsorption on Cu surfaces, in that the {100} facets are covered with halide and HDA, while the {111} facets contain only halide in a select range of solution-phase chemical potentials^{43,49,50}. Similarly, iodides can also bind strongly to Ag(111) and Ag(110), suggesting potentially a similar masking effect to that on gold⁵¹.”

Comment 4: “*The patches are primarily polymer-based. How extendable is this to other chemical ligands that may have specific roles for catalytic, magnetic, or fluorescent functions?*”

Reply: Thank you for this great comment! The patch formation and facet-selective ligand adsorption can extend to chemical ligands with specific functions. In our original manuscript, the facet-selective adsorption

starts from the thiol ligands of 2-naphthalenthioi (2-NAT), which belong to the class of small molecule functional ligands that have been often used for surface-enhanced Raman spectroscopy. While the spatial distribution of 2-NAT molecules on the particle surface is hard to see under TEM, the subsequent adsorption of PS-*b*-PAA onto the 2-NAT-coated sites helps visualize the spatially discrete “patching” of 2-NAT ligands. Note that the spatial distribution of 2-NAT is not always identical to that of PS-*b*-PAA though; the latter can be fine-tuned by the polymer chain entropy and enthalpy effects (see our reply to **Reviewer 2 Comment 1**).

To further show the generalizability of atomic stenciling to other patch compositions, we performed new experiments summarized in **Supplementary Fig. 18**, including the following. First, we used a different thiolated functional ligand, biphenyl-4-thiol (**Supplementary Fig. 18a,b**), which is often used as a negative resist for nanolithography applications (*Appl. Phys. Lett.* 75, 2401 (1999)); here PS-*b*-PAA was still added to confirm the facet-selective thiol ligand adsorption onto NP surface. Second, besides PS-*b*-PAA, we studied other block copolymers of different end functional groups, such as PS-*b*-poly(ethylene oxide) and PS-*b*-poly(acrylamide), which can also be used to form patches (**Supplementary Fig. 18c-f**).

In the revised manuscript, we have made the following changes to address the comment.

1. We have added discussions in the main text,

“The stenciling method is scalable (**Supplementary Fig. 17**) and effective for other thiols and block copolymers (**Supplementary Fig. 18**).”

2. We have added a new **Supplementary Fig. 18** to summarize the results of new ligand chemistries. The figure caption includes the details of the synthesis conditions.

Thiolated ligand variation

Block copolymer variation

Supplementary Fig. 18. Synthesis of patchy NPs with different ligands and block copolymers. (a,b) Chemical structure of biphenyl-4-thiol (a) and TEM images of patchy octahedra synthesized using biphenyl-4-thiol together with PS-*b*-PAA (b). The NPs exhibit the same vertex patches as those synthesized at the same condition (but using 2-NAT as the thiol ligands instead), which is listed in

Supplementary Table 3 (first row, with NaI). (c,d) Chemical structure of PS-*b*-poly(acrylamide) (c) and SEM images of patchy cubes synthesized using PS-*b*-poly(acrylamide) together with 2-NAT (d). (e,f) Chemical structure of PS-*b*-poly(ethylene oxide) (e) and SEM images of patchy cubes synthesized using PS-*b*-poly(ethylene oxide) together with 2-NAT (f). These NPs using different block copolymers exhibit the same face patches as those synthesized using the same condition (but using PS-*b*-PAA instead), which is listed in **Supplementary Table 6**. Scale bars: 100 nm.

Comment 5: “*Is it fair to expect thermodynamics to be the primary driving force; what about polymer chain dynamics during grafting and kinetics of assembly (e.g., diffusion of particles)? Given the high impact nature of this journal, I think it is fair for me to ask for some additional temporal (experimental) characterization to observe patch formation and at least nanoparticle assembly dynamics (again I hope I didn't miss this - the supplementary information is so large that it is easy to miss a section!); I did note the simulation movies in SI.*”

Reply: Thank you for the insightful suggestions! In the revised manuscript, on patch formation process, we have performed (1) new experiments by “quenching” the patch grafting reactions at different time points to collect the product NPs for TEM imaging (**Supplementary Fig. 37a**) and (2) new MC simulation of the patch grafting probability (**Supplementary Fig. 37b**). In both the experiment and MC simulation using gold octahedra as an example, the polymer patches are initially small and only form at a subset of the available vertices. This is associated with a kinetic effect, because chains initially can freely select any of the open six vertices for attachment, as all are favorable. By design, this will create an asymmetry in vertex coverage (i.e., not all vertices contain adsorbed chains) as there is no guarantee that free polymers in solution have sufficiently sampled all possible locations on the NP. However, with increased chain adsorption, the vertices with chains already present experience continuously increased crowding. Eventually, such crowding prohibits additional chain adsorption to the same vertex due to too high of an entropic barrier. This means that newer chains must seek out other vertices containing patches that are not large enough in size to prohibit growth or any of the previously missed (open) vertices. This process drives the evolution of the patches to become more symmetric in size and distribution, which is the thermodynamically stable state. In other words, the patch formation is mostly driven by thermodynamics, though there are kinetics-determined intermediate patch patterns. In the self-assembly of patchy NPs, the final structures observed in our experiment match with equilibrium-based MC simulation (**Figure 5**), which suggests that the self-assembly process is driven by thermodynamic equilibrium.

As to the suggested experiment to observe the particle interaction and diffusion dynamics during the self-assembly process, this could only be done using liquid-phase TEM because particle diffusions require the presence of liquids and nanometer spatial resolution in real-space and real-time. Our group has extensive expertise in liquid-phase TEM, but studying the patchy NP self-assembly dynamics by liquid-phase TEM will meet one major challenge: the superlattices are in three dimensions (3D) and liquid-phase TEM currently is two-dimensional projection-based imaging, incapable of capturing 3D structural details of each lattice layer. We acknowledge that such studies will be an exciting future direction to pursue given the enormous parameter space of directional interaction rendered by the patches. We see such efforts out of the scope of our current study.

In the revised manuscript, we have made the following changes to address the comment.

1. We have added new experiments and MC simulations on the patch formation process summarized in **Supplementary Fig. 37**.

Supplementary Fig. 37. Temporal evolution of the patch formation on gold octahedra. (a) Representative TEM images of patchy octahedra at different time points after the polymer grafting starts. Incomplete, mostly symmetry-broken vertex patches are formed initially, and small patches at all six vertices are observed at around 60 min. Subsequently, the patches gradually grow larger, and large vertex patches are observed at around 120 min. $[I^-]$ and $[2\text{-NAT}]$ are fixed at $6.62 \mu\text{M}$ and $56.7 \mu\text{M}$, respectively. (b) MC simulations of polymer chain grafting dynamics during patchy octahedra with the grafting probability p_{graft} noted. Scale bars: 20 nm.

2. We have added discussions in the main text on these new results of patch formation time series.

“We additionally note that, the patch patterning process is driven by thermodynamic equilibrium, although with kinetic formation of intermediate patterns. In both our experimental and modelling of patchy formation on octahedra, the polymer patches are initially small and only form at a subset of the available vertices (**Supplementary Fig. 37**). We associate this behaviour with kinetics. During the initial grafting, chains can freely select any of the open six vertices for attachment, as all are favourable. By design, this will create an asymmetry in vertex coverage as there is no guarantee that free polymers in solution have sufficiently sampled all possible locations on the NP. However, as more chains adsorb, the pre-occupied vertices by chains experience increased crowding, driving the patches to the equilibrium state, with more symmetric size and distribution.”

3. We added discussions on the significance of observing the self-assembly process experimentally.

“Experimental observation of the self-assembly dynamics of patchy NPs, using liquid-phase TEM⁶¹⁻⁶³ or synchrotron small-angle X-ray scattering⁶⁴, can potentially reveal how the directional interactions

govern the assembly pathways. Furthermore, extending patchy NP assembly strategies assisted by computational predictions could uncover other assembly motifs and structures.”

Comment 6: Supplementary information provides critical experimental and computational details; a minor suggestion is for the authors to also create concise summaries of the main protocols to improve readability/accessibility.

Reply: Thank you! We have the **Methods** section placed after the main text following the journal format requirement, where we summarized the protocols of patchy NP synthesis, characterization, computational and theoretical modelling. In the revised manuscript, we also added discussions of the new experiments in the **Methods** section.

Responses to Reviewer 3

Overall Comments: *“The authors report a novel way to make patchy nanoparticles of various geometric shapes (octahedron, bipyramid, cuboctahedron, cube, and rhombic dodecahedron) using an atomic, bottom-up approach analogous to macroscopic “stencilling”, that involves controlled and selective masking of gold nanoparticle facets with iodide ions followed by polystyrene-*b*-polyacrylic acid polymer brush attachment to unmasked particle areas such as vertex truncations mediated by a short hydrophobic thiol ligand (2-natphthalenethiol, 2-NAT). The authors show TEM, EDX mapping, TEM tomography and SEM images which show superb control of the polymer lobes and their positioning as a function of Iodine/2-NAT ratio. Furthermore, the authors show results from DFT, MD, MC grafting and MC simulations that both confirm and predict iodine and 2-NAT binding specificity to various Au surfaces, molecular arrangement of polymer chains, polymer patch patterns, and patchy particle assembly. Finally, patchy particles are dried using either coffee ring assembly or capillary drying and show formation of long-range ordered structures on micro to millimeter length scales.”*

“The authors present an original approach for the large-scale synthesis of patchy gold nanoparticles, that is driven by a competition of crystal lattice binding energy and entropy contributions and is potentially generalizable to other nanoparticle systems. The approach is different from previous nanoparticle patching approaches as iodine is used as a masking agent.”

“The manuscript is very well written with a concise and systematic presentation of the various steps in the patching process supporting the validity of the approach, showing a deep understanding of the mechanistic details, and an excellent command of characterization and simulation tools. The data is of superb quality and the aesthetic of the figures is very pleasing.”

Reply: We greatly appreciate all the supportive and constructive comments from the reviewer, all of which have been addressed by adding new experiments, new analysis, and new discussions as detailed below. We thank the reviewer for helping us make the work more thorough and stronger.

Comment 1: *“The authors use histograms to show patch distributions and %yield to indicate the quality of their patching process. It would be useful if the authors would define what is meant by “achieving the synthesis of more than 20 different patchy NPs in high yield (Fig1, Supplementary Figs. 3-15)” besides “simply” showing TEM images of a few particles per synthesis. Maybe the yield could be expressed in terms of the amount of gold nanoparticle seeds used and patchy particles made.”*

Comment 2: *“Page 3 Line 102, “With these two important steps demonstrated, we explore...” Please, define “high yield” in this sentence. Figure 1 and the supporting info only supply images of a few particles that show patch uniformity, but do not provide information about the yield.”*

Reply: We thank the reviewer for this great comment. The synthesis yield was calculated based on counting the number of nanoparticles (NPs) in transmission electron microscopy (TEM) and scanning electron microscopy (SEM) images, and specifically as the ratio of the number of patchy NPs of the desired patch pattern to that of all the NPs in the images. Out of the 21 types of patchy NPs that we demonstrated in our work, 17 of them have a yield higher than 80% and 9 of them have a yield higher than 90% (see statistics detailed in the newly added **Supplementary Table 1**). These results are consistent with the fact that we are able to assemble large scale superlattices using patchy NPs. In the original manuscript, **Figure 1** shows one representative particle per image to highlight the feature of patchy pattern. **Figure 3** and **Supplementary Figs. 3–15** have a combination of zoomed-out views of many NPs and representative zoomed-in images. To show more examples, we have **uploaded the raw TEM and SEM images of each of the 21 types of patchy NPs into an open-access database** with the link provided in our “Data Availability Statement”, for other researchers to study and compare as they reproduce this work.

In the revised manuscript, we have made the following changes to thoroughly address this comment.

1. We added a new **Supplementary Table 1** summarizing the details of counting methods, the statistics involved, and the synthesis yield of patchy NPs. This table also clearly describes the 21 types of patchy NPs that we demonstrated in this work.

Supplementary Table 1. Synthesis yield of patchy NPs. The yield analysis counts the ratio of the number of NPs of the desired patchy shape to the total number of NPs in the TEM and SEM images. As noted, 17 out of the 21 types of patchy NPs with distinctive patterns exhibit a synthesis yield higher than 80% (shaded in light blue). Unless otherwise noted, the core particles are made of gold.

Patchy pattern	Yield (%)	Number of the examined NP	Figure index
Octahedron 1*	94.7	57	Fig. 1 “w/ Iodide” 1st row
Octahedron 2	87.0	77	Fig. 1 “w/ Iodide” 2nd row
Octahedron 3	87.0	192	Fig. 1 “w/ Iodide” 3rd row
Octahedron 4**	86.4	110	Fig. 1 “w/ Iodide” 4th row
Bipyramid 1*	96.3	161	Fig. 1 “w/ Iodide” 1st row
Bipyramid 2	76.8	207	Fig. 1 “w/ Iodide” 2nd row
Bipyramid 3	73.6	148	Fig. 1 “w/ Iodide” 3rd row
Bipyramid 4**	90.2	132	Fig. 1 “w/ Iodide” 4th row
Cuboctahedron 1	70.8	120	Fig. 1 “w/ Iodide” 1st & 2nd rows
Cuboctahedron 2	73.5	170	Fig. 1 “w/ Iodide” 3rd & 4th rows
Cuboctahedron 3**	82.0	89	Supplementary Fig. 8 4th column
Cuboctahedron 4**	85.1	67	Supplementary Fig. 15c
Cube 1	94.2	138	Fig. 1 “w/ Iodide” 1st & 2nd rows
Cube 2	96.7	183	Fig. 1 “w/ Iodide” 3rd & 4th rows
Cube 3**	90.1	91	Supplementary Fig. 15e
Rhombic dodecahedron 1	92.9	140	Fig. 1 “w/ Iodide” 1st row
Rhombic dodecahedron 2*	96.8	93	Fig. 1 “w/ Iodide” 2nd row
Rhombic dodecahedron 3	81.8	214	Extended Data 7h–k
Rhombic dodecahedron 4**	85.6	188	Fig. 1 “w/ Iodide” 4th row
Rhombic dodecahedron 5*	80.3	183	Supplementary Fig. 13e 2nd column
Palladium cube	92.5	67	Supplementary Fig. 60m,n

* Patchy NPs with extended patches.

**** Patchy NPs with symmetry-broken structures.** Since symmetry-broken patchy NPs can have patches with varying numbers and locations, all NPs exhibiting such symmetry-broken structures are included in the count.

2. We revised the discussion on the yield in the introduction and **Figure 3** caption,

“With these two important steps demonstrated, we explore the parameter space of NP shape, iodide concentration, ligand concentration and grafting temperature, achieving the synthesis of more than 20 different patchy NPs, 17 of which are at a yield higher than 80% (**Fig. 1**, **Supplementary Figs. 3–15**, **statistics of yield analysis in Supplementary Table 1**).”

“More than 100 NPs (**d**: 208 NPs; **g**: 102 NPs; **j**: 141 NPs; **m**: 130 NPs) are analyzed to determine the synthesis yield for each patchy NP type.”

3. We have uploaded the SEM and TEM images for all the 21 types of patchy NPs listed in Supplementary Table 1 to an open-access database as the raw data associated with this paper. We added this detail in our “**Data Availability Statement**,”

“The raw TEM and SEM images of the 21 types of patchy NPs listed in **Supplementary Table 1** can be found at https://doi.org/10.13012/B2IDB-4862788_V1.”

Comment 3: “*Figure 1. It seems that w/o iodine row should come after the w/iodine row to follow the sequence in the manuscript text.*”

Reply: Thank you! We agree with the reviewer on matching the sequence of text discussion and figure component in **Figure 1**. Because the current sequence of “w/o Iodide” at the top and “w Iodide” at the bottom follows the general trend of increasing [I⁻] used in synthesis from top to bottom of the figure, we see it more appropriate to keep the sequence of the figure unchanged but adjust the text.

In the revised manuscript, we move the discussion of “w/o Iodide” before “w/ Iodide”,

“These truncations produce secondary facets (hereafter referred to as vertices), whose atomic arrangement differs from that on the primary facets (hereafter referred to as faces). Varying the reaction conditions allows control over the size, location and pattern of the patches (**Supplementary Figs. 3–15**, **Supplementary Tables 2–10**, **Supplementary Notes 2,3**). **In the absence of iodide masks, NPs are fully coated with polymers regardless of NP shape (Fig. 1c, the “w/o Iodide” region).** With iodide masks, moving left to right and top to bottom in **Fig. 1c**, we see for the octahedron and bipyramid shapes that not only is the size of the vertex patch tuneable, but the pattern transitions from all-vertex patches to a single vertex patch as the concentration of iodide is increased while keeping the 2-NAT concentration constant, breaking the inherent NP symmetry.”

Comment 4: “*Page 6 Lines 200-201, “We do observe configurations in which 2-NAT is partially...” Please, explain why it is presumed that the physisorbed 2-NAT will be washed off.*”

Reply: Thank you! In **Figure 2**, our density functional theory (DFT) calculations show clearly that with increasing iodide coverage, the Au–S distance between Au surface and 2-NAT increases (**Figure 2b**) and, for the case of the Au(111) surface, 2-NAT binding is especially weak at the highest iodide coverage (**Figure 2c**). When we decompose the total binding energy into van der Waals interactions (physisorption) and direct-bonding interactions (chemisorption), we see that 2-NAT is mostly physisorbed at the highest iodide coverage. For such weak binding, it is reasonable to assume that 2-NAT could be washed off the surface, or that it could desorb from the surface in a solution environment. In practice, the fact that the experimentally observed patchy NP patterns all match the selective chemisorption of 2-NAT due to iodide masking also confirms that the physisorbed ones are washed off and thus cannot provide a surface for polymers to be grafted onto.

In the revised manuscript, we have updated the discussions in the main text as follows:

“We do observe configurations in which 2-NAT is partially or completely physisorbed (**Fig. 2a**), but we presume that physisorbed 2-NAT would not remain on the NP surface after extensive washing nor provide stable grafting sites for polymers, because they are weakly bound (**Fig. 2c**).”

Comment 5: “Figure 2e Readability of the $\Delta\mu$ -2-NAT vs μ I- plot could be improved. Maybe it would be easier to mark the smaller fields with numbers and then put a list of the numbers and the corresponding three-color bar images in the big white space on the bottom right or maybe connect the color bars with lines to the fields they refer to.”

Reply: Thank you! In the revised manuscript, we have updated **Figure 2** and the figure caption following the reviewer’s suggestion to make it more readable:

Fig. 2 | DFT prediction of facet and concentration dependent stencilling. **a**, Geometry-optimized structures in DFT calculations of 2-NAT adsorption without (top) and in the presence of (bottom) iodide on three low-index gold facets of (111), (100) and (110) at surface coverages of 1/2 ML for iodide and 1/3 ML for 2-NAT. Aromatic rings of 2-NAT are omitted for clarity in the top panel. 2-NAT molecules chemisorbed and physisorbed on gold are coloured orange and yellow, respectively. **b**, DFT calculation of $d_{\text{Au-S}}$ with increasing iodide coverage on each facet, at a fixed 2-NAT coverage of 1/3 ML. **c**, Contributions to binding energy (E_b) of 2-NAT to each gold facet from chemisorption (grey bars) and physisorption (white bars) for a fixed 2-NAT coverage of 1/3 ML and for increasing iodide coverages from left to right. **d**, High-angle annular dark field (HAADF) imaging and EDX mapping of the iodide-incubated gold octahedron (right). The atomic ratio of Au and I in averaged line profiles across the gold octahedron face obtained from EDX (left). **e**, Phase diagram delineating equilibrium adsorption configurations on gold as a function of iodide and 2-NAT chemical potentials. The surface coverages of iodide (purple) and 2-NAT (orange) on three different facets are represented by line lengths (coverage values in **Supplementary Table 13**), corresponding to the numbered phase regions in the phase diagram. Regions in which stencil effect is predicted by DFT are shaded in cyan. Scale bars: 20 nm.

Comment 6: “Figure 4 Caption. Please, define Ω also in the caption.”

Reply: Thank you! In the revised manuscript, we have added additional descriptions of Ω in **Figure 4** caption and **Supplementary Note 6** as follows:

“Polymers of end-to-end distance R are grafted at one end onto the NP with their conformation as a function of Ω , which is a shape parameter that defines the spatially dependent surface curvature of the NP core (**Supplementary Note 6**).”

Supplementary Note 6: “ Ω is a curvature-related shape parameter that defines both the NP core geometry and position on its surface. It is defined as the value of the maximum deviation from a planar surface for each respective core NP geometry. Additionally, we select the location of the maximum curvature on the shape. For convex shapes, the maximal Ω value directly corresponds to the circumsphere-to-insphere diameter ratio.”

Comment 7: “Page 11 lines 349-353, “Usage of T ...” This sentence is not clear and could be improved.”

Reply: Thank you! In the revised manuscript, we have updated discussions as follows:

“The thermodynamic effects of the polymer chain conformation, grafted out of the available sites determined by unmasked regions, are captured in reduced coordinate $T\Omega^{1/4}$. A high $T\Omega^{1/4}$ reflects the limit where steric repulsion between neighbouring chains dominates patch patterning, while a low $T\Omega^{1/4}$ indicates that chain–chain attraction dictates the resultant polymer distribution. Notably, the symmetry-breaking limit corresponds to significant microphase separation between the PS and PAA blocks where inter-chain attraction completely dominates to break the intrinsic symmetry of iodide-masked NPs (experimental examples in **Supplementary Fig. 15**).”

Comment 8: “Page 14, Conclusion, lines 433-436. Gold seems to be rather unique in its ability to support reversible binding and not react with surface ligands or adatoms. It would be helpful if the authors could provide at least one other example system where they predict a similar approach could work. If such a system is not easy to identify, maybe the authors could at least provide guidance on what characteristics such a system would have to exhibit.”

In the revised manuscript, we have added new experiments to show the success of applying the stenciling strategy to palladium (Pd) nanocubes using iodides as masks, producing highly uniform face patches (**Supplementary Fig. 60**). The foundational principle of stenciling is to use facet-selective adsorption of species (e.g., atoms, ions, ligands) to mask surface sites to allow only the unmasked sites to be further coated. As long as this principle can work, stenciling can work for other compositions of metal NPs, though not necessarily always masking {111} or using iodides. In the case of Pd, first-principles studies using density functional theory (DFT) indicate that iodide binds more strongly to Pd(111) than to Au(111) (*Phys. Chem. Chem. Phys.* 16, 13630 (2014); *J. Electrochem. Soc.* 163, H796 (2016)) because iodide is more polarized on Pd. Moreover, Pd has a smaller lattice constant than Au, which would enhance its masking effect by steric repulsion. Relatedly, iodide can also form into dense layers on Pd(110) as discussed in Ref. (*J. Phys. Chem. C* 118, 29919 (2014)), which could preclude 2-NAT adsorption onto the Pd(110) surface.

Consistent with the above literature, both the edge Pd(110) and vertex Pd(111) facets in Pd nanocubes show no coating of polymers in our experiment (**Supplementary Fig. 60**). Iodide adsorption on Pd is generally less studied than that on Au and previous literature does not consider co-adsorption of iodide and 2-NAT as in our work. More detailed DFT analysis is needed to fully understand the mechanism of stenciling for other metal NPs or stencils, which goes beyond the scope of this first study.

In the revised manuscript, we have made the following revisions to address this comment.

1. We added a new **Supplementary Fig. 60** summarizing our experiment of making Pd patchy nanocubes.

Supplementary Fig. 60. Synthesis of patchy palladium nanocubes. (a) Schematic illustrating the synthesis of palladium cubes. Iodides are introduced during the palladium nanocube synthesis, intrinsically enabling selective facet masking for polymer grafting. (b) EDX mapping of the palladium cubes showing composition uniformity. (c–l) High-angle annular dark field (HAADF)-STEM images of as-synthesized palladium cubes and corresponding 3D schematic. (d,i) show the viewing direction of the STEM images. (e–g) and (j–l) are zoomed-in views of the NP regions boxed in (c) and (h), showing that the vertices, edges, and faces exhibit $\{111\}$, $\{110\}$, and $\{100\}$ facets, respectively, as noted by the yellow lines. (m,n) Representative SEM images of patchy palladium cubes. Inset: TEM image (m) and corresponding 3D schematic (n) of a patchy cube with cyan colored region as polymer patches. For these samples, iodides exist as an additive during the palladium cube synthesis process as shown in a. The as-synthesized NPs are used directly for polymer grafting without additional iodide masking step. (o,p) Representative SEM image of patchy palladium cubes synthesized with the additional iodide masking step performed on the as-synthesized palladium cubes (o) and SEM image with selected particles false-colored to highlight the patches (cyan). As more iodide is added, the patch sizes decrease, consistent with the general trend of increased masking at higher $[I^-]$ during stenciling (as seen from m,n to o,p). See **Supplementary Note 2.8** for synthesis details. Scale bars: (b) 50 nm, (c,h) 20 nm, (e–g, j–l) 2 nm, and (m–p) 100 nm (inset: 20 nm).

- We added a new **Supplementary Note 2.8** on the experimental details of patchy palladium nanocube synthesis.

“2.8. Synthesis of patchy palladium nanocubes

Palladium nanocubes are synthesized following a previously reported method with slight modifications⁷. 20 mM palladium(II) chloride (H_2PdCl_4) is first prepared by dissolving 11.78 mg of

sodium tetrachloropalladate(II) (Na_2PdCl_4) in 2 mL of 40 mM HCl. The solution is tightly capped and kept undisturbed in a water bath at 30°C for 1 h. Then, 625 μL of the prepared 20 mM H_2PdCl_4 is added to 6.25 mL of 100 mM CTAB in a 20 mL vial, followed by sequential addition of 1.25 mL of 40 mM potassium iodide (KI) 0.5 mL of 100 mM AA, and 0.75 mL of DI water under shaking at 400 rpm. The vial is capped and quickly transferred to an oil bath, followed by heating at 90°C for 1 h with stirring at 400 rpm. Afterward, the reaction solution is cooled by immersion in a water bath. To remove unreacted reactants, the as-synthesized palladium cube solution is transferred into a 15 mL centrifuge tube and centrifuged twice at 8,500 rpm for 10 min each. After the first centrifugation, the supernatant is removed, and the pellet is redispersed in 5 mL of DI water. Following the second round, the supernatant is removed again, and the pellet is dispersed in 2.5 mL of 20 mM CTAB.

Patchy palladium cubes shown in **Supplementary Fig. 60m,n** are synthesized following the same general procedure as described in Supplementary Note 2.1, but with adjusted [2-NAT] and without the iodide masking step. Specifically, 213 μL of the palladium cube solution from above is transferred into a 1.5 mL microcentrifuge tube and centrifuged at 5,500 rpm for 15 min. The 10 μL pellet is redispersed in a 15 mL centrifuge tube using 20 mM CTAC to reach a total volume of 7.2 mL. The solution is then centrifuged twice. After the first centrifugation at 7,600 rpm for 15 min, the pellet is redispersed in 10 mL of 20 mM CTAC and centrifuged again at 7,600 rpm for 15 min. After the second centrifugation, the pellet is redispersed in 1 mL of DI water and transferred into a 1.5 mL microcentrifuge tube, followed by a final centrifugation at 3,250 rpm for 15 min. After removing the supernatant, the final CTAB concentration and volume are adjusted to 0.07 mM and 610 μL , respectively, using additional DI water and 0.16 mM CTAB solution. For polymer grafting, 815 μL of DMF is first added to an 8 mL vial. Then 5 μL of 2-NAT solution (0.02 mg/mL in DMF), 200 μL of the adjusted palladium cube solution, and 80 μL of PS-*b*-PAA solution (8 mg/mL in DMF) are sequentially mixed by dropwise addition under mild vortex. The vial is tightly capped with a Teflon-lined cap, sonicated for 5 s, sealed with parafilm, heated at 110 °C in an oil bath, and left undisturbed for 2 h. The reaction mixture is then cooled down to RT in the oil bath, which typically takes 90 min. The solution is transferred to 1.5 mL centrifuge tubes and centrifuged three times at 4,500 rpm, 2,250 rpm, and 1,750 rpm for 15 min each, to separate the residual 2-NAT and PS-*b*-PAA from the patchy Pd NPs. After the first and second centrifugations, 1.49 mL of the supernatant is removed, and the 10 μL pellet is redispersed with 1.49 mL of water. Following the third round, the 10 μL pellet is diluted with 490 μL of water for long-term storage for TEM observations. Patchy palladium cubes shown in **Supplementary 60o,p** are synthesized the same as above, except that the as-synthesized palladium nanocubes undergo an additional iodide masking step with a final [NaI] concentration of 6.62 μM .”

3. We added additional discussions on the new experiments and more literature in the main text.

“It can also be applied to other metal NPs such as palladium nanocubes (**Supplementary Fig. 60**), where the presence of iodide leads to the formation of face-patched palladium nanocubes, similar to the face-patched gold nanocubes. Iodide adsorption on Pd(111) and Pd(110) has been discussed in previous literature^{47,48}, though the specific 2-NAT co-adsorption mechanism needs further study. The general principle of facet-selective adsorption and masking is likely to be generalizable to other systems. For example, DFT calculations and experiments show a potential masking effect of iodide or chloride for hexadecylamine (HDA) adsorption on Cu surfaces, in that the {100} facets are covered with halide and HDA, while the {111} facets contain only halide in a select range of solution-phase chemical potentials^{43,49,50}. Similarly, iodides can also bind strongly to Ag(111) and Ag(110), suggesting potentially a similar masking effect to that on gold⁵¹.”

Comment 9: “*Extended Data Fig 2. The variability of the gold nanoparticle shapes in this figure is very surprising and requires at least a comment as to why this data is shown. The synthesis seems to involve nanorods followed by etching processes leading to very uniform gold nanoparticle seeds as shown in the supporting info, but ED Figure 2 shows the often-encountered variability in gold nanoparticle shapes.*”

Reply: We thank the reviewer for the great comment! As the reviewer correctly pointed out, we employed core NP synthesis involving nanorod etching, which provides uniform NP seeds and polyhedral NPs as shown in **Supplementary Figs. 2** and **48**. Importantly, in our standard patchy NP synthesis, the shape uniformity of core gold NPs is retained after iodide masking and polymer grafting, as supported by UV-Vis spectra (**Extended Data Fig. 5a**), the high uniformity observed in our large-scale self-assemblies (**Figure 5**), and the yield analysis (**Supplementary Table 1** and our reply to **Reviewer 3 Comments 1 and 2**).

The variations of core gold NPs in **Extended Data Fig. 2** results from a special sample preparation for cutting-edge electron microscopy imaging of surface adsorption of submonolayers of iodide, which requires exceptionally clean samples. Specifically, iodide-masked gold NPs were extensively washed to reduce cetyltrimethylammonium bromide (CTAB) concentration to the micromolar level. After depositing the sample onto the TEM grid, it was fully dried, plasma cleaned, and baked overnight at 130°C under high vacuum to remove residual CTAB. We suspect that this rigorous CTAB removal left gold surfaces unprotected, leading to instability and occasional NP merging, especially when they sat too close to each other on the grid. For the energy dispersive X-ray analysis, we focused on the intact octahedra. Note that by the time the NPs are on the TEM grid, we do not expect the fact of iodide adsorption to change (although the detailed spatial distribution could change due to particle surface restructuring) since the NPs are no longer in a solution or in any form of iodide stock.

In the revised manuscript, we have updated the **Extended Data Fig. 2** caption with explanations,

“The reduced shape uniformity of gold NPs in this sample is presumably due to the extensive CTAB removal (see **Methods**) required for high-sensitivity STEM-EDX detection of submonolayer iodide, which may destabilize unprotected gold surfaces. For EDX line profile analysis, we focus on intact octahedra oriented in the [110] projection (highlighted with yellow dotted circles).”

Comment 10: “*It is somewhat surprising that the assembly experiments are only shown for the most symmetrically modified patches. Although this may be outside the scope of this already very dense paper, curiosity drives the following questions: Have the authors tried assembly of the non-symmetrically patched nanoparticles? Were difficulties encountered in these systems? Or do the calculations predict interesting structures for asymmetric patch geometries?*”

Reply: Thank you! In our original manuscript, we showed an example of the self-assembly structure from non-symmetrically patched cuboctahedra in **Supplementary Fig. 59a**. While the assembly exhibits some features reminiscent of the body-centered cubic ordering seen in symmetrically face-patched cuboctahedra, the degree of local ordering is low. We suspect this is due to asymmetric patch configuration disrupting the directional and uniform multivalent interactions that promote long-range order, through repeating unit cell symmetry. Additionally, the larger exposed gold surfaces on non-symmetrically patched cuboctahedra may reduce electrostatic repulsion between NPs (originated mostly from charged patches), leading to formation of kinetically trapped aggregates. This trend is also consistent with the case shown in **Supplementary Fig. 59b**, where octahedra with symmetrically arranged but smaller vertex patches do not assemble into highly ordered structures either, likely due to insufficient surface coverage by polymer patches. We note that other self-assembly strategies, such as liquid-liquid interfacial assembly and solution-phase assembly, can potentially overcome these challenges.

In the revised manuscript, we have made the following revisions to address this comment.

1. We added new schematics for NP patch patterns and revised captions in **Supplementary Fig. 59**.

Supplementary Fig. 59. Patch symmetry and size effect on the patchy NP self-assembly. (a) SEM images of self-assemblies from asymmetrically patched cuboctahedra featuring hybrid face and vertex patches on the same particle (a) (particles prior to assembly: **Supplementary Fig. 8**, $[I^-] = 0.83 \mu\text{M}$). Local order is disrupted, likely due to asymmetric patching and reduced electrostatic repulsion from larger exposed gold surfaces. (b) SEM images of self-assemblies from octahedra with symmetrically arranged but smaller vertex patches (b) (particles prior to assembly: **Supplementary Fig. 3**, $[I^-] = 117.6 \mu\text{M}$), which fail to assemble into ordered structures, possibly due to insufficient polymer coverage typically required for long-range ordering (inset: corresponding 3D model for (a) the example of symmetry-broken patchy cuboctahedra and (b) patchy octahedra with small patches). For detailed synthesis conditions of these patchy NPs, see **Supplementary Tables 2,5**. Scale bars: 500 nm.

2. We added additional discussions in the main text regarding asymmetrically patched NP assemblies.

“Notably, in addition to the robustness of stencil-induced patchy NP assemblies, the patch **pattern symmetry**, size, and thickness are crucial for long-range, open structure formation. **The assembly of asymmetrically patched NPs shows disrupted local ordering.** Furthermore, if the patches are too small or too thin relative to the masked facets, the strongly attractive gold–gold van der Waals forces between all facets can lead to kinetic trapping, resulting in poorly ordered structures (**Supplementary Fig. 59**).”

“Furthermore, extending patchy NP assembly strategies assisted by computational predictions could uncover other assembly motifs and structures.”

Thank you all for your great comments and suggestions! We are happy that all the referees regarded their comments addressed in our last revision. We have done further editing of the manuscript to address the editorial comments on the formatting, without altering the scientific descriptions and conclusion of the work.

Referee #1:

Remarks to the Author: *“The authors have adequately addressed my comments. I now fully endorse the publication of the manuscript as it is.”*

Reply: Thank you for all your constructive and insightful comments to make our work stronger! We are happy that you are supportive of publishing the manuscript as it is.

Referee #2:

Remarks to the Author: *“I think the authors have satisfactorily addressed the reviewer comments. I recommend this interesting publication.”*

Reply: Thank you for all your constructive and insightful comments to make our work stronger, as well as finding our work interesting!

Referee #3:

Remarks to the Author: *“The authors have done an excellent job in addressing the various questions and comments from the three reviewers. The manuscript has been improved further and is ready for publication.”*

Reply: Thank you for all your constructive and thorough comments on making our work stronger. We are happy that you appreciate our revision and support the publication of this work.